# LCice 1.0: A generalized Ice Sheet Systems Model coupler for LOVECLIM version 1.3: description, sensitivities, and validation with the Glacial Systems Model (GSM version D2017.aug17)

Taimaz Bahadory[1] and Lev Tarasov[1]

[1]Dept. of Physics and Physical Oceanography, Memorial University of Newfoundland, St. John's, NL, Canada

*Correspondence to:* Lev Tarasov (lev@mun.ca)

**Abstract.** We have coupled an Earth System Model of Intermediate Complexity (LOVECLIM) to the Glacial Systems Model (GSM) using the LCice 1.0 coupler. The coupling scheme is flexible enough to enable asynchronous coupling between any glacial cycle ice sheet model and (with some code work) any Earth system Model of Intermediate Complexity (EMIC). This coupling includes a number of interactions between ice sheets and climate that are often neglected: dynamic meltwater runoff routing, novel downscaling for precipitation that corrects orographic forcing to the higher resolution ice sheet grid ("advective precipitation"), dynamic vertical temperature gradient, and ocean temperatures for sub-shelf melt. The sensitivity of the coupled model with respect to the selected parameterizations and coupling schemes is investigated. Each new coupling feature is shown to have a significant impact on ice sheet evolution.

An ensemble of runs is used to explore the behaviour of the coupled model over a set of 2000 parameter vectors using Present-Day (PD) initial and boundary conditions. The ensemble of coupled model runs is compared against PD reanalysis data for atmosphere (2 meter temperature, precipitation, jet-stream and Rossby number of jet), ocean (sea ice and Atlantic Meridional Overturning Circulation (AMOC)), and Northern Hemisphere ice sheet thickness and extent. The parameter vectors are then narrowed by rejecting model runs (1700 CE to present) with regional land ice volume changes beyond an acceptance range. The selected sub-set forms the basis for ongoing work to explore the spatial-temporal phase space of the last two glacial cycles.

## 1   Introduction

Transitions between glacial and interglacial states have been a periodic feature of the Earth's climate for the last few million years. The driver of these transitions is understood to be orbital forcing (Berger, 2014; Birch et al., 2016; Birch et al., 2017; Hahn et al., 2015; Rind et al., 1989), with an important role for $CO_2$ variations (Elison Timm et al., 2015; Ganopolski et al., 2016). Nevertheless, the role that climate feedbacks play in amplifying or inhibiting the responses to these forcings is not clear. Given available proxy data, how well do we know the progression of these glacial cycles? Is there more than one way each transition could have occurred? How sensitive were these glacial cycles to small perturbations in the external forcings (*e.g.,* volcanic eruptions)? In order to address these questions, we need to understand the relative importance of different feedbacks

between ice sheets and other aspects of the climate system. We can build such understanding by probing this phase space with physically-based models that include the pertinent feedbacks on glacial timescales.

Temperature and net precipitation (the solid/liquid fraction thereof) encompass the main atmospheric impacts on ice sheets. Marginal ice sheet surface mass-balance is very sensitive to the vertical temperature gradient. As indicated by the observations presented by Gardner et al. (2009), the vertical surface temperature gradient ("slope lapse rate") can be significantly different from the free-air temperature lapse-rate over the Greenland ice sheet. Furthermore, neither of these vertical temperature gradients are *apriori* appropriate for downscaling near surface temperatures to a higher horizontal resolution grid. The actual vertical gradient required is that due to changing the surface topography in the climate model. However, most coupled model studies use a fixed vertical temperature gradient set to an approximate mean free-air lapse-rate (usually between 5 to 7 K/km) to downscale surface temperatures from coarse climate model grids (Glover, 1999; Flowers and Clarke, 2002; Thomas et al., 2003; Arnold et al., 2006; Bassford et al., 2006b, a; De Woul et al., 2006; Raper and Braithwaite, 2006). An approach somewhat more self-consistent with the atmospheric component of the climate model is provided by Roche et al. (2014). Their coupler extracts the vertical "along-slope surface temperature gradient" from the atmospheric model and uses it to downscale temperatures to the ice sheet model. They find this dynamic approach has significant impacts, especially over mountainous regions and Greenland.

Coarse grid climate models used in long-time integrations can not resolve surface slopes on the generally much higher resolution ice sheet grids to which they are coupled. Given the strong impact of orographic forcing on precipitation, this can potentially introduce large errors in surface mass balance, especially near ice sheet margins and over rough topography. Standard bi-linear interpolation schemes for downscaling precipitation to the ice sheet grid in turn preserve these errors.

Ice sheets directly affect the atmosphere via changing land surface type (affecting albedo, surface roughness, and moisture fluxes) and changing topography. Upscaling of topography from the relatively high resolution grids of ice sheet models to the course resolution atmospheric grids (especially for fast glacial cycle context models) has a range of options between conserving peak heights and mean heights. There is no clear criteria for a "best" choice and the sensitivity to this choice is generally unclear.

Ice-sheets primarily affect oceans directly through meltwater runoff, and changing ocean bathymetry and landmask (especially gateways). The effect of ice sheet runoff on the ocean, especially the AMOC, has been the focus of many studies Timmermann et al. (2003); Rahmstorf et al. (2005); Stouffer et al. (2006); Krebs and Timmermann (2007); Hu et al. (2008); Otto-Bliesner and Brady (2010); Kageyama et al. (2013); Xun et al. (2013); Roberts et al. (2014). Their findings show the modelled AMOC is a function of the models and coupling procedures used, in addition to the initial and boundary conditions of the experiments. These experiments generally include prescribed freshwater discharge fluxes into the ocean, in part to isolate AMOC sensitivity to freshwater forcing. The feedback of the resulting climate response on ice sheet discharge is therefore absent.

The strongest direct impact of the oceans on ice sheets is submarine melt of tide-water glaciers and sub-ice shelf melt. However, for continental scale coupled models, sub-shelf melt is either completely ignored (Ridley et al., 2005), or parameterized in a highly simplified way, *e.g.,* (Roche et al., 2014).

In this study, our objective is to develop a coupled ice sheet - climate model which encompasses most relevant feedbacks/interactions between the cryosphere and the atmosphere and ocean for continental glacial cycle scale contexts. Through a selection of ensemble parameters, we are also working towards bracketing the strength of these feedbacks across model ensembles. We also examined sensitivity to coupling time-step by setting up three similar simulations with different coupling time-steps (100, 20, and 10 years). Features of note in the coupling we describe herein include:

1. Dynamic vertical 2 meter temperature gradient to improve the temperature downscaling from the atmosphere model to the ice sheet model.

2. An advective precipitation downscaling scheme which accounts for wind velocity and topographic slopes.

3. Dynamic meltwater routing.

4. An efficient scheme to extract approximate lat/long gridded ocean temperature fields from LOVECLIM ocean temperature profiles for sub ice shelf melt computation

Table 1 compares the interactions between ice sheets and climate models only infrequently included in previous coupled modelling studies to this one. There are two main interactions yet to be implemented. First, the dust cycle and its impact on atmospheric radiative balance and ice surface albedo (and therefore surface mass balance) awaits future work. Second, the LOVECLIM ocean component does not handle changing bathymetry and landmask over a transient run. It does have a parameterized Bering Strait throughflow which permits shutdown of throughflow when local water depth approaches zero.

Climate models used for glacial cycle contexts need to be fast enough to simulate tens of thousands of years in a reasonable time interval, while sufficiently complex to include all important climate dynamics. We tested every freely available fast model that included ocean, atmosphere and dynamical sea ice components, and found a number of published models to be numerically unstable or otherwise unable to run or port. The only stable model with all these components was LOVECLIM. The other models tested and associated porting failures are:

**SPEEDO** : compilation error using PGI and Intel compilers.

**FOAM (v. 1.5)** : no dynamic sea ice model; compilation error using PGI and Intel compilers.

**OSUVic (v. 2.8)** : compilation error.

**CSIRO-Mk3L (v. 1.2)** : compilation error using PGI, Intel, and GCC compilers; problem accessing fftw library.

The paper is structured as follows. We first introduce the models in section 2. Next, we describe the coupling schemes between the ice sheet model and the atmosphere and the ocean models in section 3. In this section, we use the last glacial inception timeframe (120 - 110 ka) to show that inclusion of each process coupling scheme can have significant impact on the evolution of major NH ice sheets. In section 4, we introduce our chosen set of ensemble parameters for the coupled model. In order to justify this choice of ensemble parameters, we examine the sensitivity of the coupled model to changes in each parameter for PD climate. Then we sieve the ensemble parameter set using our coupled model with historical/PD initial and boundary conditions via a comparison against observational/reanalysis data.

**Table 1.** Feedbacks/interactions sporadically included in previous studies between the ice sheet model and the rest of the climate system, compared to the current study. None include changes to land mask and bathymetry except for parameterized Bering Strait throughflow.

| Source | Advective precipitation | Dynamic vertical temperature gradient | Dynamic meltwater runoff routing | Sub-shelf melt | Dust deposition |
|---|---|---|---|---|---|
| Stokes et al. (2012) | ✗ | ✗ | ✗ | ✗ | ✗ |
| Yin et al. (2014) | ✗ | ✗ | ✗ | ✗ | ✗ |
| Roche et al. (2007) | ✗ | ✗ | ✓ | ✗ | ✗ |
| Galle et al. (1992) | ✗ | ✗ | ✗ | ✗ | ✗ |
| Ganopolski et al. (2010) | ✗ | ✗ | ✗ | ✗ | ✓ |
| Roche et al. (2014) | ✗ | ✓ | ✗ | ✓ | ✗ |
| Heinemann et al. (2014) | ✗ | ✗ | ✗ | ✗ | ✗ |
| Current work | ✓ | ✓ | ✓ | ✓ | ✗ |

## 2 Models

### 2.1 LOVECLIM

LOVECLIM (version 1.3) is a coupled Earth Systems Model of Intermediate Complexity (EMIC), which consists of atmosphere (ECBilt), ocean with dynamic sea ice (CLIO) and vegetation (VECODE) modules. It is fast enough to simulate the last glacial inception (120 ka to 100 ka) in less than 3 weeks using a single computer core. Therefore, it has been used to simulate a wide range of different climates from the last glacial maximum (Roche et al., 2007) through the Holocene (Renssen et al., 2009) and the last millennium (Goosse et al., 2005) to the future (Goosse et al., 2007).

**Atmosphere**  The atmospheric component (ECBilt, Opsteegh et al., 1998) is a spectral global quasi-geostrophic model, with T21 truncation, three vertical layers at 800, 500 and 200 hPa, and a time-step of 4 hours. The quasi-geostrophic structure of the model limits its ability to simulate equatorial variability and hence atmospheric interactions between the tropics and higher latitudes. To partially compensate, it has additional ageostrophic terms to improve the representation of Hadley cell dynamics (Opsteegh et al., 1998). Precipitation is computed from the precipitable water of the first layer according to a precipitation threshold for relative humidity (default 85%). The model contains simple schemes for short and long wave radiation, with radiative cloud cover prescribed by default (Haarsma et al., 1996).

**Ocean**  The oceanic component (CLIO: Coupled Large-scale Ice Ocean) is a 3D primitive equation model with Boussinesq and hydrostatic approximations. The model is discretized horizontally on a $3° \times 3°$ Arakawa B-grid, with 20 vertical levels on a z-coordinate. This coarse resolution enables CLIO to run fast enough for glacial cycle simulations. A free surface and

a parameterization of down-sloping currents (Campin and Goosse, 1999) enables CLIO to receive freshwater fluxes and capture some of their impacts on dense water flows off continental shelves. Goosse et al. (2001) describe the model in detail. A major limitation of this model (and challenge for many GCMs) for paleoclimate studies is that the bathymetry and land mask can't be changed during a transient run (specifically, there is no available nor described implementation that can do so).

**Sea ice** The sea ice component of CLIO is an updated version of the Fichefet and Maqueda (1997) dynamic-thermodynamic sea ice model. A visco-plastic rheology (Hibler, 1979) is used for horizontal stress-balance. The thermodynamic component of the sea ice model considers sub-grid sea ice and snow cover thickness distribution, and ice and snow sensible and latent heat storage.

**Vegetation** VECODE is a dynamic terrestrial vegetation model with simplified terrestrial carbon cycle (Brovkin et al., 2002). The model simulates the dynamics of two plant functional types (trees and grasses), in addition to deserts, and evolves their grid-cell fractions. These fractions are determined by the contemporaneous climate state and terrestrial carbon pool. More details about the model can be found in Brovkin et al. (1997) and Brovkin et al. (2002).

LOVECLIM has been tested for both interglacial and glacial contexts. Nikolova et al. (2013) found that the large-scale changes in climate simulated by LOVECLIM for the last interglacial were in approximate agreement with that indicated by available proxies (differences in the $\pm$ 2.5 °C range for summer, and -5 to 0 °C in winter). These changes were also similar to that from a full-complexity atmosphere-ocean general circulation model (CCSM3). However due to stronger polar amplification in LOVECLIM, smaller sea ice extent and higher surface temperatures are simulated in LOVECLIM compared to CCSM3 for the interglacial.

During the Last Glacial Maximum, LOVECLIM overestimates both the minimum and maximum southern hemisphere sea ice cover compared to paleo-proxy data, while CCSM only overestimates the minimum sea ice extent (Roche et al., 2012). Roche et al. (2007) also found a reasonable agreement between the atmospheric and oceanic estimates of LOVECLIM and proxy data during the LGM (*e.g.,* disappearance of much of PD Siberian boreal forest, seasonal sea-ice extent, sea surface temperature). However, the Atlantic deep ocean circulation is stronger in their simulation, which opposes the general inference of weaker AMOC during the LGM.

## 2.2 GSM

The GSM is built around a thermo-mechanically coupled ice sheet model. It includes a 4 km deep permafrost-resolving bed thermal model (Tarasov and Peltier, 2007), fast surface drainage and lake solver (Tarasov and Peltier, 2006), visco-elastic bedrock deformation (Tarasov and Peltier, 1997), Positive Degree Day surface mass balance with temperature dependent degree-day coefficients derived from energy balance modelling results (Tarasov and Peltier, 2002), sub-grid ice flow and surface mass balance for grid cells with incomplete ice cover (Morzadec and Tarasov, 2017), and various ice calving schemes for both marine and pro-glacial lake contexts (Tarasov et al., 2012). For the results herein, ice shelves are treated using a crude shallow ice approximation with fast sliding. The GSM is run at 0.5°longitude by 0.25°latitude grid resolution.

The GSM has three new features that haven't been previously documented. First, ice calving has been upgraded to the more physically based scheme of DeConto and Pollard (2016). However, our implementation imposes the additional condition that the ice cliff failure mechanism is only imposed at ice marginal grid cells. Second, a temperature-dependent sub-shelf melt scheme that also depends on adjacent sub-glacial meltwater discharge from the grounded ice sheet has been added. The melt is proportional to the water temperature to the power 1.6 and to proximal sub-glacial meltwater discharge following the

Greenland fjord modelling results of Xu et al. (2013). We also impose a quadratic dependence on ice thickness to concentrate sub-shelf melt near deep grounding lines in accord with the results of process modelling (*e.g.,* Jacobs et al., 1992). Finally, a first order approximation to geoidal deflection is now included. Details of these schemes will be in an upcoming submission fully describing the revised GSM.

## 2.3    Model initialization

For the results herein, PD and glacial inception model runs are initiated with PD ice sheet thickness. The initial bed-thermal temperature field is set to the PD resultant field from a mix of best-fit past calibrated and ongoing calibration model runs (*i.e.,* without LOVECLIM as in Tarasov et al. (2012) for North America).

To initialize the temperature field of existing ice sheets, results from previous transient runs are usually used in the GSM. However, this doesn't work for a cold start from PD fields, as the PD ice thickness fields won't line up fully with model results.

As such, the ice temperature field is initially linearly interpolated from surface temperature to a basal temperature of -3°C. This enforces a frozen base to insure a smooth spin-up but otherwise provides warm enough ice to generate significant ice velocities. Ice velocity fields are then computed. The ice thermodynamics is subsequently partially spun-up over 5000 years (with fully coupled bed-thermal evolution). When the results of pre-Eemian Greenland and Antarctic calibrations become available, this will be used to initialize the respective Eemian ice sheet temperature and ice thickness fields.

Glacial inception surface topography is also offset from PD by the amount required to remove PD topographic discrepancies (to observed) from some past best fit calibration model runs.

As detailed below, the climate model is spun-up over an ensemble parameter dependent time interval prior to onset of the coupled model run.

## 3    Coupling

The coupler is designed to regrid and exchange data between the ice sheet model (GSM) and LOVECLIM (ECBilt and CLIO) in both directions with minimal adjustment to the model code. Fig. 1 displays all the fields the coupler transfers between different component models and the processes involved.

Due to the computational costs of coupling and trivial variations of ice sheets over small time-scales, the ice model and the climate model run for a certain number of years before receiving updated fields from the other model. On the other hand, using large coupling time-steps can also introduce errors into the results. To test the effect of the coupling time-step on ice sheet evolution, we used three different coupling steps (100, 20, and 10 years) to simulate the last glacial inception starting at 120

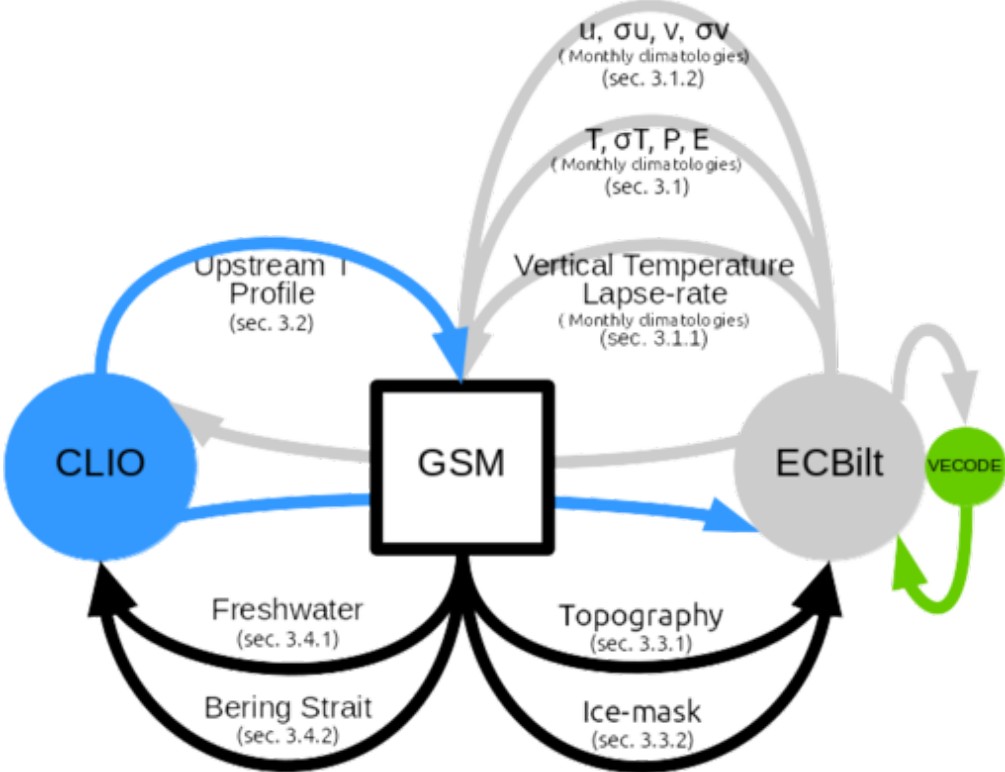

**Figure 1.** Components of the climate system and interactions between them included in the coupled model, with the section numbers in which each process is described in detail. Atmospheric fields passed from ECBilt to the GSM are monthly climatologies.

ka. With identical boundary and initial conditions for all three simulations, runs with 10 and 20 year coupling steps have less than a maximum of 3% difference in ice volume (Fig. 2). The 100-year coupling-step run (red line in Fig. 2), however, strongly diverges from the other two during the retreat phase. This ice volume divergence is mostly due to a thinner ice in North America (NA) and Eurasia (EA), and a less southern extent of the NA ice sheet. A weaker response with longer coupling time-steps is expected given the delay in updating climate and ice boundary conditions for the GSM and LOVECLIM respectively. Given these results, we choose 20 years as the coupling step for all of our ensemble simulations (in part given the not insignificant overhead with the coupler as currently coded/scripted).

The ice sheet model exchanges data with both the atmosphere and the ocean models at the end of each coupling step. The fields that are passed are described in detail below.

### 3.1 Atmosphere to ice

At the end of each coupling time-step, the coupler receives climate fields averaged over the last 10 years from ECBilt and converts them to monthly-mean values. These fields include:

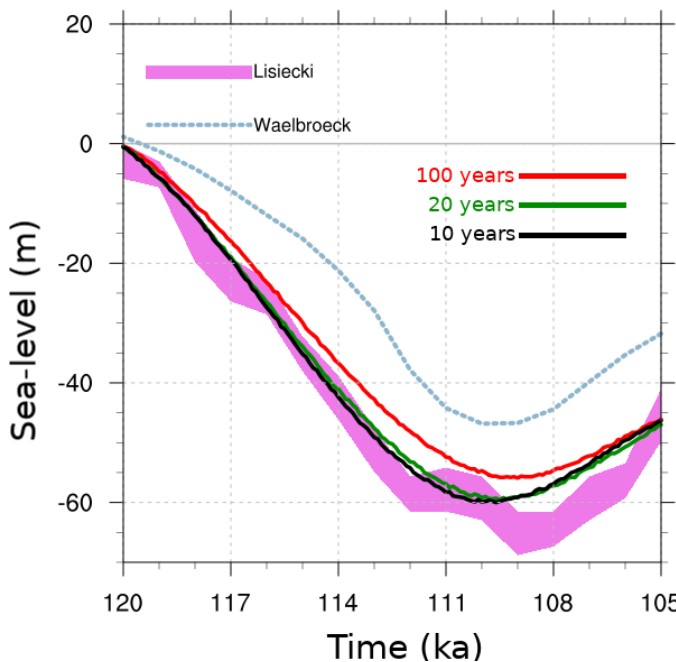

**Figure 2.** Total ice volume in sea level equivalent (m) at last glacial inception, coupled synchronously with 100, 20 and 10 year time-steps.

- 2 m near surface air temperature and standard deviation

- Vertical 2 m temperature gradient

- Precipitation

- Evaporation

- Latitudinal and longitudinal components of wind and standard deviation of each

LOVECLIM computes both 2 m near surface air temperatures (T2m) and surface (or skin) temperatures. We note at least one previous study indicates usage of LOVECLIM surface temperature for ice sheet modelling contexts (Roche et al., 2014), which we find problematic. Surface melt determination using positive degree days requires the former and rain/snow fraction determination will be more accurately estimated with the former than the latter. Ice thermodynamics would properly use the latter but this can be alleviated in part if the ice sheet model limits surface temperatures to 0°C over ice and snow for the ice thermodynamics.

Given the simplified boundary layer physics of LOVECLIM, it may be that some weighted average of its T2m and surface temperature is a more appropriate estimate of "true" 2 meter temperature. As shown in the supplement, a raw average gives somewhat better overall fits to ERA40 2 m temperatures over Greenland and Antarctica but worse fits for July over North America and especially Eurasia using the default LOVECLIM tuning. Given these mixed results (and the possibility that after

retuning the average of T2m and surface temperature would give better fits), we provide an option in the coupler to extract this average temperature from LOVECLIM instead of T2m.

    The large difference in spatial resolution of the two models necessitates horizontal and vertical downscaling of the climatic fields. The GSM receives climatic fields on the LOVECLIM grid, and downscales them to its own grid resolution using bi-linear interpolation.

The downscaled standard deviation of temperature (using 4-hourly ECBilt data for each month averaged over the last 10 years of each coupling time-step) is used to compute monthly Positive Degree Days, with the usual assumption of a Gaussian distribution around the monthly mean. This is opposed to the traditional practice of assuming a constant value, usually between 5°C and 7°C.

### 3.1.1   Vertical temperature gradient

Large grid resolution differences between ECBilt and the GSM result in surface elevation differences between the two models, especially in places with steep topography. The altitude dependence of temperature in such regions can drastically affect the type of precipitation and surface mass balance of the ice sheet. Therefore, in addition to horizontally downscaling the temperature from LOVECLIM to the GSM, a vertical correction of temperature is required.

    By monitoring 25 sites spread over a 15,650 km$^2$ area and with an altitude range of 130 to 2010 m on the Prince of Wales
Icefield for two years, Marshall et al. (2007) found a mean daily vertical surface temperature gradient of -4.1 K/km, with an average summer gradient of -4.3 K/km. These values are less than the standard mean free-air temperature lapse-rate that is often used for extrapolations of sea level temperature to higher altitudes (-6.5 K/km) (*e.g.,* Glover, 1999; Arnold et al., 2006; Raper and Braithwaite, 2006). Marshall et al. (2007) also find a vertical surface temperature gradient of -6 to -7 K/km on steep regions in summer, and around -2 K/km in regions where northerly anticyclonic flow is more common. In addition, Gardner
et al. (2009) find significant spatio-temporal variations in vertical temperature gradients across four glaciers in the Canadian high Arctic.

    The GSM uses the near-surface vertical T2m gradient calculated by the coupler at the end of each time-step to downscale the temperature field over its high resolution grid. In each LOVECLIM grid cell, the coupler first determines the highest and lowest elevations from the GSM topography constrained by the cell's boundary. Next, the T2m for these two elevations
is calculated using the inherited scheme from the LOVECLIM atmospheric model (as detailed in Roche et al., 2014). The resulting temperatures and elevation difference between the two points is then used to calculate the temperature lapse-rate in that LOVECLIM grid cell.

    Fig. 3.a and 3.b show the present-day vertical T2m lapse-rate calculated by the coupler for summer and winter. The derived lapse-rate has strong spatial and temporal variation over NA and Greenland. The impact of this variation is shown in Fig. 3.c. Starting from the same 110 ka configuration, the difference in ice thickness after 2 kyr between a dynamic temperature lapse-rate run and a control run (default LOVECLIM parameters) with 6.5 K/km lapse rate can reach over 1 km.

    Evaluating the appropriateness of our vertical temperature downscaling approach is difficult, especially when considering glacial/interglacial changes. Using a global climate model (CCSM3), Erokhina et al. (2017) found significantly larger surface

5   slope lapse-rate values over the Greenland ice sheet during LGM compared to pre-industrial values (February mean increase of about 3.7 K/km and about 0.9 K/km for July). In contrast, our T2m mean LGM lapse-rate over Greenland is $0.8 K/km$ stronger for February and $0.2 K/km$ weaker for July compared to that of PD. However, neither lapse rate is *apriori* an accurate choice for vertical downscaling. A need remains for a multi-resolution modelling study to compare a "true" downscaling vertical temperature gradient with the various possible lapse-rates that can be derived from a single resolution model.

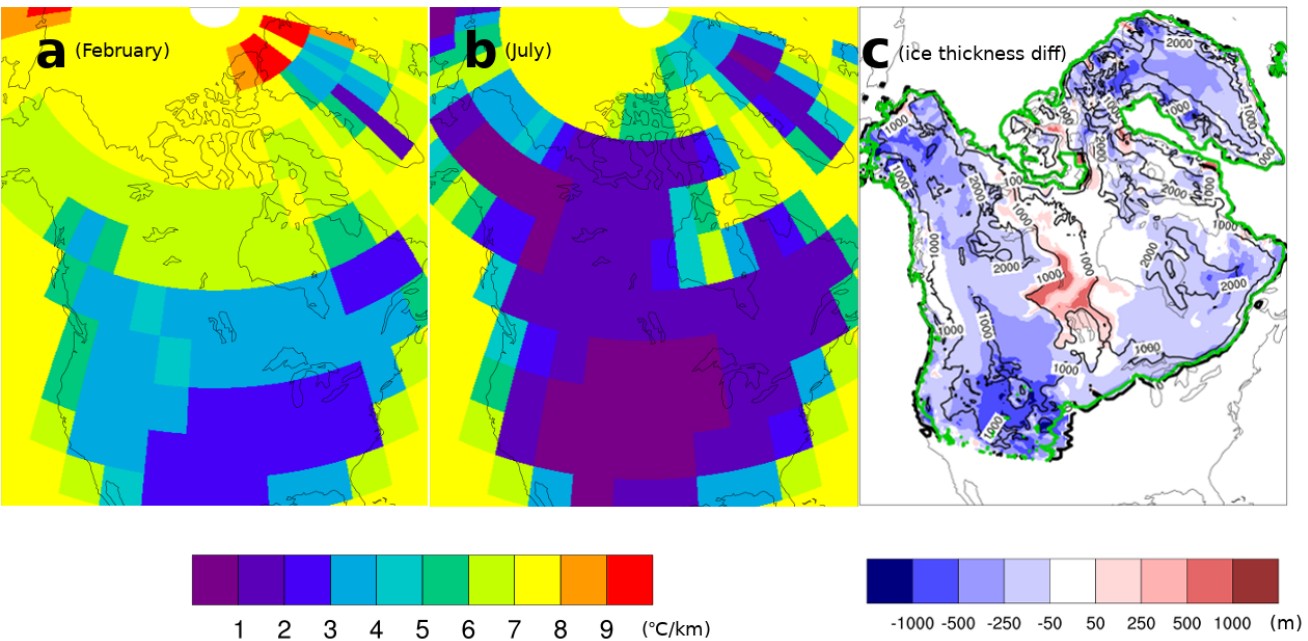

**Figure 3.** Vertical temperature lapse-rate calculated by the coupler at PD over NA in **a**. February, and **b**. July. **c**. shows the ice thickness difference between dynamic and constant 6.5 K/km lapse rate (control) runs after running for 2 kyr, starting from the same 110 ka configuration. Black contours show the ice thickness in the control run. Thick black and green contours show the ice margin in the control and dynamic lapse-rate run, respectively.

### 3.1.2   Advective precipitation downscaling

LOVECLIM calculates evaporation, rain, and snow for each grid cell based on its coarse resolution surface topography and temperature fields. These fields require downscaling to the higher resolution GSM grid. A common approach is to linearly interpolate both precipitation and evaporation fields onto the high resolution ice sheet model grid, calculate the net precipitation amount, and finally determine the amount of rain and snow for each grid cell using the downscaled temperature. However, linear interpolation does not correct the damped orographic forcing due to a coarse resolution climate model grid. Here, we apply a new approach to precipitation downscaling that also accounts for orographic forcing at the ice sheet grid resolution.

5   The scheme assumes that orographic precipitation effects for upslope winds will be proportional to the vertical velocity induced by the surface slope and therefore to the dot product of the horizontal wind velocity and surface slope ($S_{GSM}$ and

$S_{ATM}$) with the latter given by:

$$S_{GSM}(\boldsymbol{u}(\boldsymbol{x},\boldsymbol{y},\boldsymbol{month},\boldsymbol{k})) = \boldsymbol{u} \cdot \nabla h, \qquad S_{ATM}(\boldsymbol{u}(\boldsymbol{x},\boldsymbol{y},\boldsymbol{month},\boldsymbol{k})) = \boldsymbol{u} \cdot \nabla h_{ATM} \tag{1}$$

The $k$ in the above equations indexes a representative range of wind vectors $(\boldsymbol{u}(\boldsymbol{x},\boldsymbol{y},\boldsymbol{month},\boldsymbol{k}))$ for each month. To simplify the coupling and still capture wind variation, we use monthly climatologies of mean wind velocity and its standard deviation in the determination of $S_{GSM}$ and $S_{ATM}$. We compute the advective precipitation correction factor ($fk_p$) using in turn the $S$'s as a function of mean, and mean $\pm$ one standard deviation and then sum over these factors with appropriate weights ($W(k)$) for a Gaussian distribution. This correction is based either on the ratio of the $S$ terms for $S_{ATM} > 0$ (*i.e.,* upslope winds) or else their difference (to transition into precipitation-shadowing). In detail, with the inclusion of a regularization term ($\mu$, that governs the transition to precipitation-shadowing) and bounds ($f_{pmin}$ and $f_{pmax}$), this takes the form:

$$k = 1, 3$$

$$S_{ATM}(x,y,month,k) > 0 : fk_p(x,y,month,k) = MIN\left[MAX\left(\frac{S_{GSM}+\mu}{S_{ATM}+\mu}, f_{pmin}\right), f_{pmax}\right] \times W(k)$$

$$S_{ATM}(x,y,month,k) \le 0 : fk_p(x,y,month,k) = MIN\left[MAX\left(\frac{S_{GSM}-S_{ATM}+\mu}{\mu}, f_{pmin}\right), f_{pmax}\right] \times W(k)$$

The loop is carried out for each point on the GSM grid. The net correction for each corresponding point on the lower resolution atmospheric grid is then accumulated to generate a rescaling coefficient that is mapped back to each $f_p(x,y,month)$ on the GSM grid to ensure mass conservation. The scheme is currently implemented with $\mu = 0.005$ and $f_{pmin} = 0.2$ and $f_{pmax} = 5.0$.

The new advective precipitation downscaling results in increased ice sheet volume and southern extent for the North American ice sheet during the inception phase. This increase is largest for the southeastern sector of the ice sheet (Fig. 6). Ice thickness also decreases in some regions due to precipitation-shadowing.

### 3.1.3 No bias correction

Studies of Goosse et al. (2007); Mairesse et al. (2013); Renssen et al. (2009); Widmann et al. (2010); Roche et al. (2007); van Meerbeeck et al. (2009); Otto-Bliesner and Brady (2010) demonstrate LOVECLIM's overall ability to simulate last millennium, Holocene, and the Last Glacial Maximum (LGM) climates in agreement with observed and proxy records. However, the model still suffers from a high temperature bias at low latitudes, a too-symmetric distribution of precipitation between the two hemispheres, an overestimation of precipitation and vegetation cover in the subtropics, weak atmospheric circulation, and an overestimation of the ocean heat uptake over the last decades (Goosse et al., 2010).

The extent to which these biases are due to the tuning of LOVECLIM parameters and missing couplings with the rest of the Earth/climate system is unclear. We therefore do not apply a bias correction to atmospheric fields and instead examine the extent to which an ensemble parameter sweep can reduce the bias. As detailed below, a reduction in PD regional temperature and precipitation bias occurs for various members of our perturbed parameter ensemble.

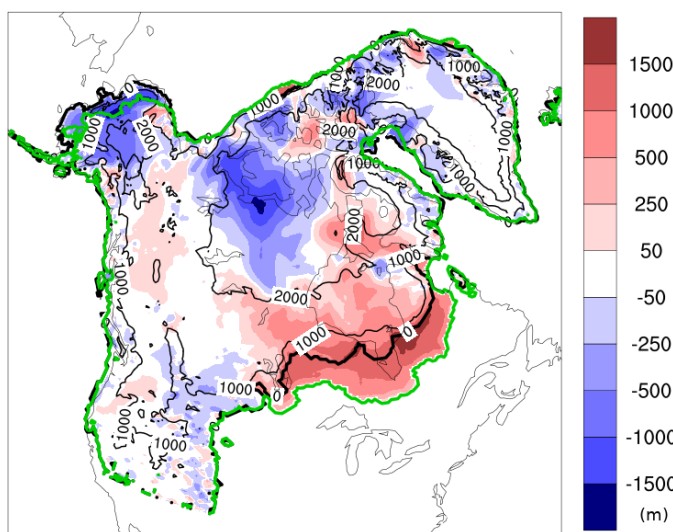

**Figure 4.** Impact of advective precipitation downscaling inclusion in the coupled model; NA ice thickness difference at 110 ka between simulations with and without the advective precipitation method. Contours show the ice thickness in the control run. Thick black and green contours show the ice margin in the control and advective precipitation run, respectively. LOVECLIM parameters are set to default values.

The control run (with all LOVECLIM parameters set to their default values, and other coupling parameters as described in the caption of Table 2) shows the highest temperature bias in the "Southern NA" region ($\sim 5°$C), with slightly colder temperatures in the "North NA" ($\sim 1°$C). The temperature bias over EA is less significant, and is also less latitude dependent (both "Northern EA" and "Southern EA" biased by $< 2°$C). A reduction in regional temperature and precipitation bias is observed in various members of our later introduced ensemble of simulations for PD. The regional temperature and precipitation bias relative to observed (Table 3) over NA and EA can reach zero for some ensemble members for both summer and winter. Although there is no individual run with zero bias in all the regions, a number of selected runs show reduced temperature biases (between -1°C and 1°C) in all the four regions compared to that of the control run.

## 3.2 Ocean to ice: sub-shelf melt

Sub ice shelf melt is a challenge for paleo coupled ice sheet climate modelling given the dependence on unresolved basin-scale circulation. As a first order approximation, we assume that upstream ocean temperature at the same depth corresponds to the local sub-shelf temperature. To facilitate fast and simplified coupling, given the complexity of ocean grids in most ocean general circulation models, we only extract upstream ocean temperature vertical profiles from LOVECLIM at the end of each coupling time-step for a number of chosen index sites as indicated in Fig. 5 and use these for downstream marine sectors. We selected these sites (7 over NA+Greenland and 4 over EA) by examining PD ocean temperature climatologies from CLIO (at various depths) while taking into account ocean currents. Our site selection was predicated on the constant bathymetry and land mask of CLIO and would need updating for a model with dynamic landmask/bathymetry. The downstream masks for the

profile sites extend onto land where applicable when grounding line retreat beyond the fixed ocean mask of CLIO (*i.e.,* onto the land mask) is possible.

To test the impact of this regional disaggregation of ocean temperatures, we generated three test cases: ocean temperature forcing set to PD value, to -2°C, and set to the contemporaneous average across the above index sites. Starting from a 110 ka restart, all three options have local ice thickness differences greater than 1 km after 2 kyr compared to that with the standard coupling.

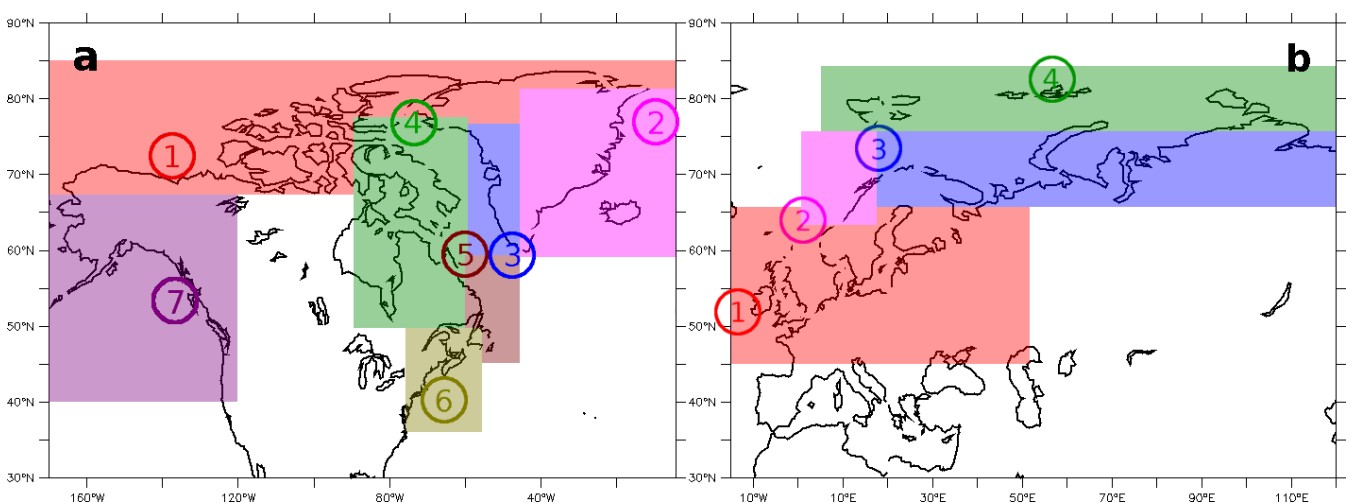

**Figure 5.** The upstream ocean temperature profile sites and corresponding downstream sectors assigned to these profiles for ocean-ice coupling in **a**. NA and Greenland, and **b**. EA.

## 3.3 Ice to atmosphere

Changes in both the topography and the ice-mask can affect the global circulation patterns by influencing the stationary waves and the jet-stream. At the end of each coupling time-step, the coupler receives the updated topography and ice thickness fields from the GSM. The topography field is upscaled to the ECBilt grid and then used for the next LOVECLIM run step. Given the large difference in grid resolution, the choice of upscaling scheme is not *a priori* clear. We have therefore implemented three different schemes to upscale the topography from the GSM high resolution grid to the ECBilt low resolution grid.

### 3.3.1 Topography upscaling and ice-mask

**Simple average method** In this method, the coupler simply calculates a weight for each high-resolution grid cell based on the fraction of the cell located inside the coarse grid cell. These weights are then used to calculate the average altitude of each ECBilt cell from the GSM orography.

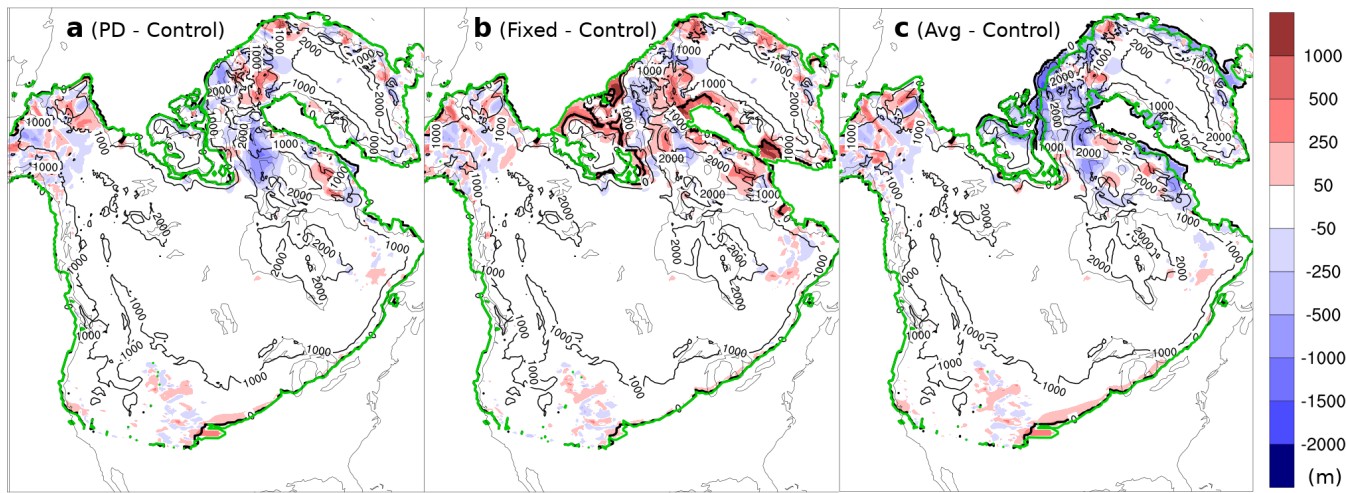

**Figure 6.** Ice thickness difference at 110 ka from the control run (dynamic ocean temperature) for: **a**. PD ocean temperature run, **b**. fixed ocean temperature at -2°C run, and **c**. temperature averaged over ocean layers run. Contours show the ice thickness in the control run. Thick black and green contours show the ice margin in the control and the other run, respectively.

**Envelope method** In the envelope method, a weighted standard deviation of the altitude of all the GSM cells inside the ECBilt cell is added to the simple average altitude from the previous method. The envelope method works reasonably well to preserve the overall topographic peaks, but it can introduce a phase shift in the terrain field, broaden ridges, and raise the height of even relatively broad valleys.

$$H_{i,j} = \overline{H_{i,j}} + \omega \times \sigma_{i,j}$$

Here, $H_{i,j}$ is the model terrain height, $\omega$ is a predefined weighting factor (in our experiments 0.5), and $\sigma_{i,j}$ is the standard deviation at the model grid point.

**Silhouette method** The silhouette method combines the simple average altitude with a silhouette height. The silhouette height is defined as follows:

$$H_s = \omega_1 H_{max} + (1 - \omega_1) \times \frac{(H_{sx} + H_{sy})}{2}$$

where $H_{max}$ is the maximum height of all GSM grid cells inside the ECBilt cell, $H_{sx}$ and $H_{sy}$ are the average peak heights obtained in each row and column of nested cells, and $\omega_1$ is the predefined weight. The silhouette height is then used to calculate the ECBilt cell altitude using:

$$H = \omega_2 H_s + (1 - \omega_2) \times H_{mean}$$

Different combinations of weighting factors $\omega_1$ and $\omega_2$ will draw the gridded terrain analysis toward preserving the peaks ($\omega_2 = 1$) or preserving the mean topographic height ($\omega_2 = 0$), which allows a greater degree of freedom to determine the model terrain analysis.

### 3.3.2 Ice-mask

Another important consideration in the ice sheet - atmospheric coupling is the variation in ice extent, which changes the albedo calculated by ECBilt and, hence, affects the temperature field over the region and globally. We used the ice thickness field generated by the GSM to create the ice-mask needed by ECBilt. To do so, the high resolution ice thickness field is first regridded to the ECBilt coarse resolution grid by using one of the methods mentioned above. Any cell in the resulting grid with more than 30% ice coverage is then assumed to be ice covered.

Our choice of a 30% threshold (as opposed to say 50%) was motivated by the following logic. For any atmospheric grid cell covering an ice margin segment, the temperature passed to the GSM should most importantly reflect ice covered boundary conditions local to the ablations zone of the ice sheet. Allowance for subgrid advection of warmer air masses from adjacent ice-free land somewhat tempers this logic. Given potentially significant impacts on critical ablation temperatures and therefore ice-sheet mass-balance, this ice-fraction threshold deserves a sensitivity analysis (in future work).

### 3.4 Ice to ocean

#### 3.4.1 Topographically-self-consistent and mass conserving freshwater discharge

The melting of continental ice sheets provides a freshwater source to the ocean that affects global sea level and the AMOC. Dynamical ocean models indicate that the strength of the AMOC in the North Atlantic Ocean is sensitive to the freshwater budget at the sites of formation of North Atlantic Deep Water (Rahmstorf, 1995).

As the GSM self-consistently computes surface drainage (while conserving mass) for the evolving topography while LOVE-CLIM surface drainage is hard-coded for PD topography, precipitation within LOVECLIM is masked out where covered by the GSM grid. The coupler then distributes GSM freshwater ocean discharge to corresponding LOVECLIM ocean discharge grid cells. Where the GSM grid edge is terrestrial, the GSM discharge is added to the corresponding LOVECLIM grid cell for internal runoff routing. As topographic gradient changes from Glacial Isostatic Adjustment (GIA) are small outside of the GSM grid, this scheme should give results close to what would be achieved with a drainage solver using a topography globally subject to GIA.

We describe the runoff routing in the coupler and the connection between the GSM and LOVECLIM drainage basins in more detail in the supplement.

Fig. 7a provides an example of how the inclusion of meltwater runoff in the coupled model improves ice sheet growth at glacial inception. Although the impact is small at the first stages of ice formation due to small ice volumes with negligible runoff rate changes, ice in the simulation including runoff grows faster as it gains more volume. The difference reaches its maximum at 110 ka, with about 50% more ice in the run with dynamic drainage routing, including much thicker and more extensive ice over North America (Fig. 7b). Compared to the control run (no dynamic drainage routing), the AMOC strength drops by 15%, and the sea ice extent shows an increase of 5% in winter and ∼15% in summer (not shown). The combination of these AMOC and sea ice changes yields a cooler summer in the run with dynamic drainage routing, and hence less ice sheet melt.

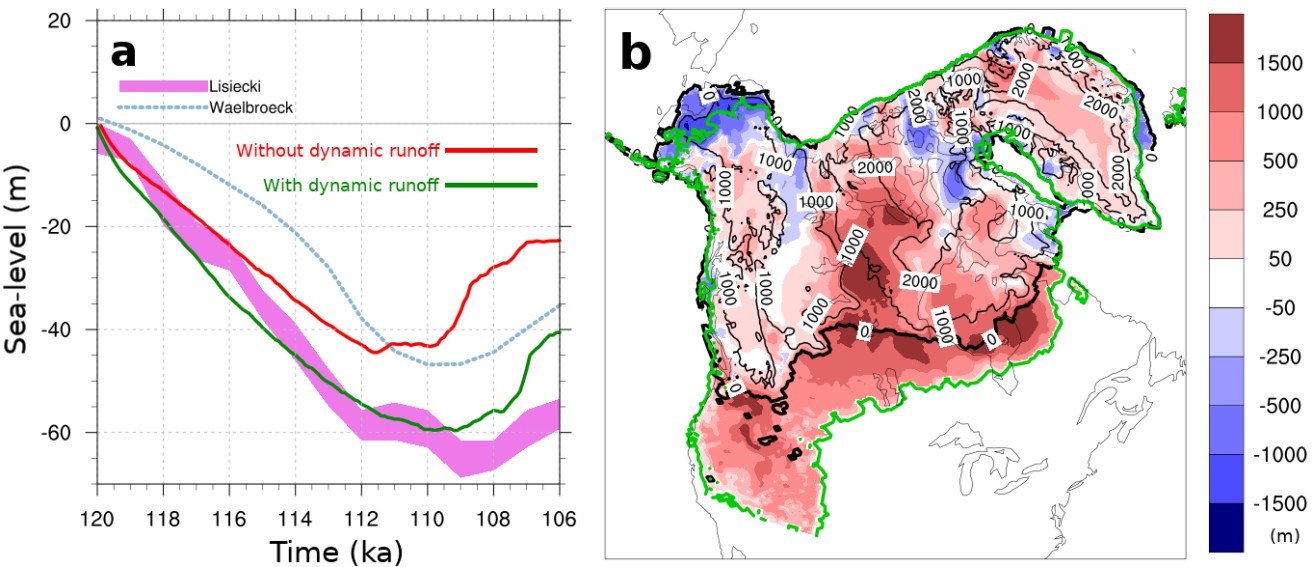

**Figure 7.** Impact of meltwater runoff inclusion in the coupled model; **a**. Total ice volume evolution at glacial inception with (green) and without (red) dynamic meltwater routing, and **b**. NA ice thickness difference at 110 ka with and without dynamic runoff routing. Contours show the ice thickness in the simulation without dynamic runoff routing. Thick black and green contours show the ice margin in the control and dynamic runoff routing run, respectively.

### 3.4.2 Bering Strait

The Bering Strait is a narrow strait with a present depth of approximately 50 m between Siberia and Alaska, through which relatively fresh North Pacific water is transported to the Arctic. From there, the North Pacific water is transported to the Greenland Sea and North Atlantic. This less-saline water affects the upper ocean stratification and thus the strength of deep ocean convection and the AMOC, which in return, has impacts on the global climate (Shaffer and Bendtsen, 1994; De Boer and Nof, 2004; Hu et al., 2008).

Due to the ice sheet growth and associated sea level lowering, the Bering Strait was often closed during glacial cycles, limiting the freshwater flow from the Pacific Ocean to the Arctic. LOVECLIM does not explicitly compute the direct connection between the Pacific and the Arctic through the Bering Strait, so the transport is parameterized by a linear function of the cross-strait sea level difference in accordance with geostrophic control theory (Goosse et al., 1997). The coupler interpolates the Bering Strait scaling at each coupling step between the PD value (0.3, 50 meter depth) and a closed strait (0.0, 0 meter depth) using the relative sea level at the Bering Strait as computed by the GSM. Given the shallowness of the strait, the accuracy of the GSM in representing sea level changes (given its viscoelastic bedrock response and first order Geoidal correction) has a potentially important role here.

## 4   Ensemble parameter sensitivity analysis

Ensemble parameters were initially chosen by judgement of their control of a physical aspect of ice sheet - climate interaction
(*e.g.,* albedo) or by their potential impact on the coupling between the ice sheet and the climate (*e.g.,* upscaling method) (Table
2). This choice was then validated by the following sensitivity analysis.

As the context of the model development is glacial inception and deglaciation, we are interested in the ensemble performance
for climate metrics which control the growth and decay of the NH ice sheets during these two stages. Therefore we use summer
2 meter temperature and winter precipitation over land. To enable comparison against observations, our sensitivity analysis is
based on transient runs over the historical interval (up to 1980 CE).

Since glacial inception and deglaciation are triggered at different latitudes in NA and EA, we have divided each continent
into diagnostic north and south zones (called "NorthNA", "SouthNA", "NorthEA", and "SouthEA"). The sensitivity of the
coupled model is tracked for each individual zone. The "NorthNA" and "SouthNA" zones cover latitude ranges of 65-75°N and
40-60°N over NA, respectively. "NorthEA" and "SouthEA" are defined over 70-80°N and 55-70°N latitude bands, respectively.
The regional boundaries are illustrated in Fig. 8.

The third column in Table 2 shows the sensitivity of T2m and precipitation in the coupled model to changes in each param-
eter through its range for four different latitudinal bands over NA and EA averaged over the 1950-1980 interval. For easier
comparison, all figures use the same temperature and precipitation scales. The sensitivity to each parameter for the four re-
gions is different for temperature and precipitation. For instance, switching between PD radiative cloud forcing and cloud
parameterization strongly affects both temperature and precipitation over all regions, while changing the snow albedo has its
strongest impact on EA temperatures and precipitation (Table 2). Each of the ensemble parameters has an impact of at least
4°C on temperature and/or 1 cm/month on precipitation over the given parameter ranges. We take this as justification for their
continued use as ensemble parameters.

In the following subsections, we further describe the parameters used in the ensemble simulation. Later, we will show the
chosen set of ensemble parameters is adequate for bracketing the relevant (temperature and precipitation ) fields of the climate
system.

### 4.1   Snow and ice albedo

Changes in the snow and ice area and type have an amplifying effect on climate by modifying the surface albedo. During
summer, the balance between absorbed and reflected solar energy at the ice sheet surface is the dominant factor controlling
surface melt variability in the ablation zone (van den Broeke et al., 2008). The parameterization of the surface albedo in
LOVECLIM takes into account the state of the surface (frozen or melting) and the thickness of the snow and ice covers
(Goosse et al., 2010). We include all types of snow and ice albedo (*i.e.,* snow, melting snow, and bare ice) in our ensemble
parameter set.

The "Snow Albedo", "Bare Ice Albedo", and "Melting Ice Albedo" rows in Table 2 show the range of albedo values for each
type and their climate sensitivity. Increasing snow albedo results in a reduction of winter precipitation over all four regions

**Table 2.** Ensemble parameters that are varied in the historical transient simulation ensemble. Column 2: the distribution of parameter values versus their range for each parameter. Column 3: change in 1950-1980 mean summer 2 meter temperature and winter precipitation over four selected regions when each parameter is varied independently from its minimum to its maximum value. In each sensitivity run, all the other parameters are fixed to LOVECLIM default values, with spin-up length: 4000 years, simple upscaling method, default LOVECLIM cloud radiative forcing, start-year: 1500 AD, and dynamic vertical temperature lapse-rate.

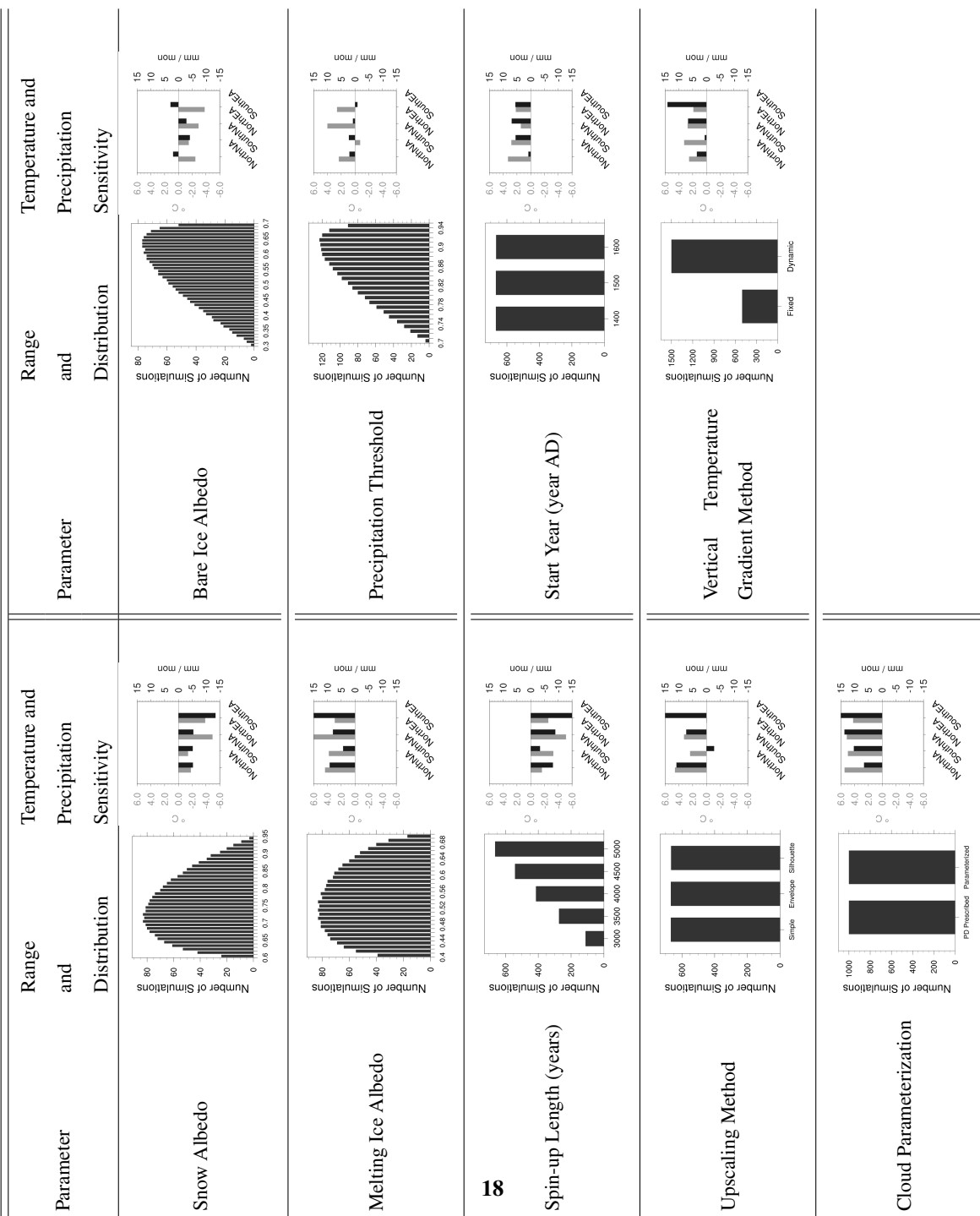

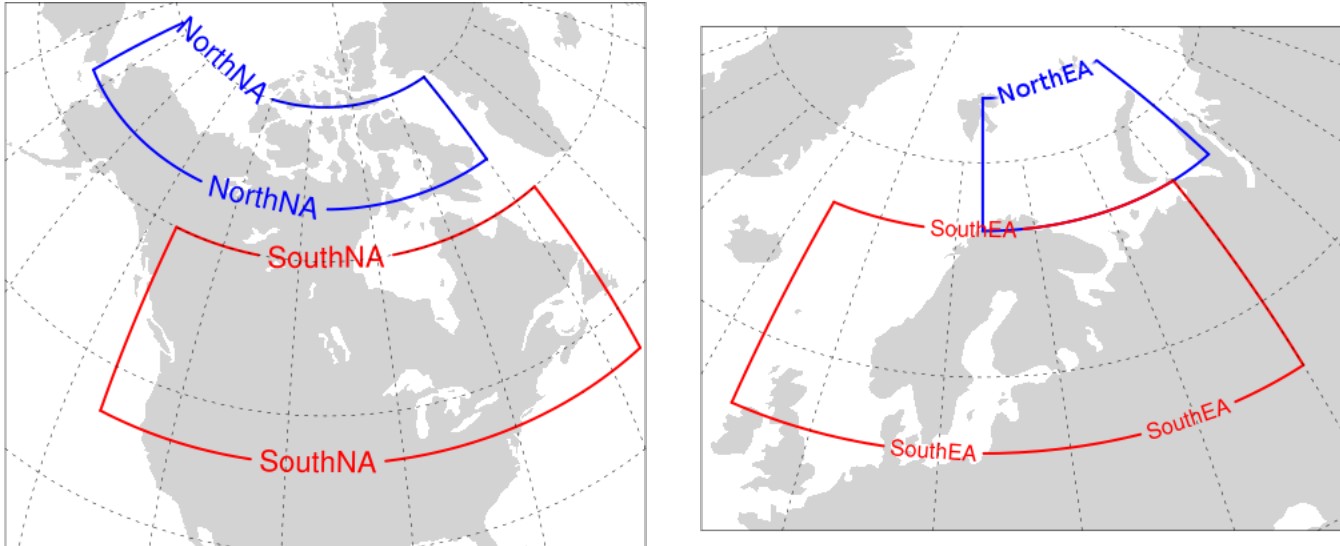

**Figure 8.** Selected north and south zones over North America and Eurasia for PD sensitivity analysis.

with an extended effect over summer temperatures, as expected. However, the same feedback is not as straightforward for bare ice albedo and melting ice albedo. Although the increase in bare ice albedo shows an expected cooling effect over all regions
10  with a smaller influence on precipitation, an increase in melting ice albedo causes all regions to get warmer between 4 and 6°C and increases the winter precipitation.

## 4.2  Climate initialization/spin-up

Before starting a coupled transient climate simulation, it is necessary to allow the atmosphere and ocean to adjust to the initial boundary conditions and external forcings. Model spin-up, and therefore the initial state of the climate system, can be a major
15  source of uncertainty in climate modelling especially given the millennial timescale of deep ocean circulation.

The general approach to spin-up the ocean is to run the ocean to an equilibrium state under fixed external forcings (*e.g.,* Manabe et al., 1991; Johns et al., 1997). However, as the climate system is unlikely to ever be in equilibrium, this choice lacks justification. We include two parameters to control the initial state of the system: LOVECLIM spin-up start year, and LOVECLIM spin-up length. All spin-ups are performed using transient orbital and $CO_2$ forcings ranging from 3000 to 5000 years but without the GSM coupling. The combination of these two spin-up control parameters results in slightly different coupled transient start times, each with different initial ocean and atmosphere states. For the runs herein, we constrain the
5  spin-up to end between 1400 and 1600 CE. Increasing the spin-up length has a cooling and drying effect in the coupled model with PD boundary and initial conditions, while starting the transient coupled run from earlier years results in slightly warmer and wetter conditions (Table 2).

## 4.3 Upscaling

The three different upscaling methods described in section 3.3.1 are evenly distributed between ensemble members as shown in Table 2. By switching between three methods, we calculate the highest temperature and precipitation changes over four regions and plotted in the last column of Table 2. The highest temperature sensitivity to the upscaling method is recorded in NorthNA followed by the NorthEA zones, and the highest precipitation sensitivity is seen in the SouthEA zone.

## 4.4 Precipitation threshold

ECBilt accounts for humidity, and thus precipitable water, only between the surface and the 500 hPa layer. Above 500 hPa, the atmosphere is assumed to be dry, so all the water transported by atmospheric flows into this region precipitates. Below the 500 hPa layer, ECBilt precipitates all the excess water above a fixed threshold (default 0.83) multiplied by the vertically integrated saturation specific humidity (Goosse et al., 2010). This parameter has the largest relative impact for NorthEA temperature (4°C over the parameter range equivalent to 60% of mean).

## 4.5 Cloud radiation parameterization

The representation of clouds is one of the largest sources of uncertainty in models. They play an important role in regulating the surface energy balance of ice sheets, with competing warming and cooling effects at the surface through changes to short- and long-wave radiative fluxes. The effect of ice sheets on cloud formation is also significant. The growth of ice sheets results in tropospheric cooling and a reduction in humidity. This colder and drier troposphere displaces the upper tropospheric stratiform clouds downward, and reduces the low level stratiform cloud cover around the ice sheets (Hewitt and Mitchell, 1997).

The total downward and upward long-wave radiative scheme in ECBilt is a function of the vertical profile of the temperature, the concentration of various GHGs, and the humidity, and is computed for both clear-sky and cloudy conditions. The radiation computed for each grid cell is then the weighted average of these two conditions based on the cloud coverage. The default ECBilt configuration prescribes radiative cloud coverage to the PD ISCCP D2 data-set (Rossow, 1996). The total downward and upward shortwave radiative fluxes depend on the transmissivity of the atmosphere, which also relies on the prescribed cloud cover (Goosse et al., 2010).

Given the importance of cloud radiative feedbacks on ice sheet evolution, the use of a prescribed PD cloud cover for paleoclimate modelling lacks justification. Therefore, we have added a simple cloud parameterization scheme similar to the precipitation parameterization scheme as described in section 4.4. The only difference here is the humidity threshold for cloud formation, which is assumed to be 10% less than the precipitation threshold, allowing cloud cover without precipitation. Including the dynamic cloud cover radiation feedback in the coupled model slightly decreases the total ice volume at glacial inception (Fig. 9) through reduced humidity during glacial conditions reducing the cloud cover.

As evident in the "Temperature and Precipitation Sensitivity" column in Table 2, regional temperature and precipitation is sensitive to each of our ensemble parameters in the coupled model. However, due to the non-linearity of the climate system,

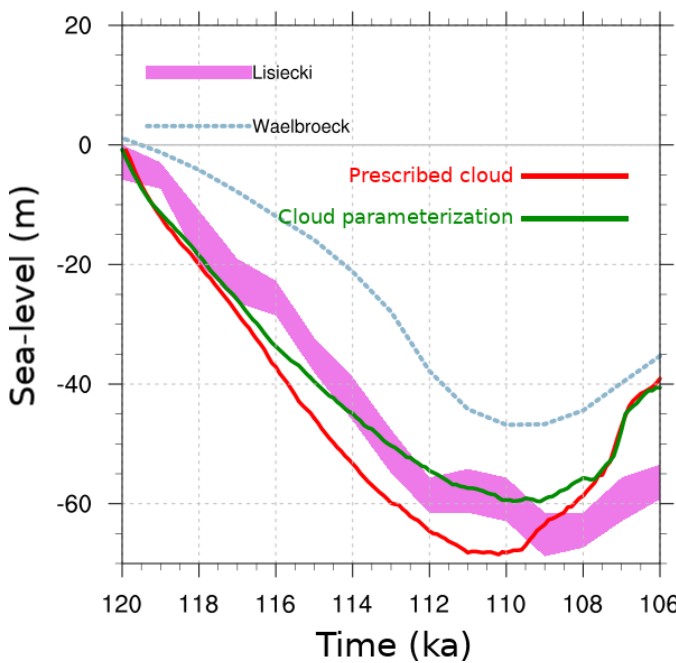

**Figure 9.** Total ice volume at the last glacial inception with the cloud parameterization (green) and with the PD cloud cover forcing (red).

the combined effect can be significantly different. In the next section, we explore the coupled model response to all these parameters in an ensemble of simulations.

## 5   PD ensemble results

The fast runtime of the coupled model permitted an initial ensemble of 2000 PD simulations using the fully-coupled GSM-LOVECLIM and varying the model parameters described above. We chose the PD interval to permit comparison of the coupled model output against observational data and to select a better fit sub-ensemble for transient paleo runs. All simulations are spun-up using transient forcings (orbital (Berger, 1978) and GHG Law Dome (for recent CO2 data) and Dome C (Etheridge et al., 1998; Monnin et al., 2001) (for pre-industrial to 5 ka)) for 3000 to 5000 years without the GSM coupled, followed by a transient coupled run ending at year 1980. The ensemble parameter values were generated via a Latin-Hypercube scheme with increased weighting near LOVECLIM default values.

Given a priority to "bracket reality" and limitations of the component models, we chose to not use climate characteristics for the sub-ensemble filter. Our focus on coupled ice and climate and our choice to avoid bias corrections led to a trial criteria based on ice volume changes (between 1700 and 1980 CE). Therefore we used the PD simulated NH ice sheet growth to sieve out parameter vectors with major surface mass-balance biases. We first considered a less than 0.1 m SLE change in ice volume requirement for each of the 3 northern ice sheet regions. But this already fell below our target size of 500 simulations

with NA being the problematic region (Fig. 10). So we changed the criterion to be the 500 runs with the least amount of ice volume change over each ice sheet. The sub-ensemble NA simulations have ice volume less than 0.15 m SLE and ice volume changes for the other two ice sheets well below 0.1 m SLE (Fig. 10). Crucial to our "reality bracketing", there are about 80 sub-ensemble members with ice loss over the given time interval.

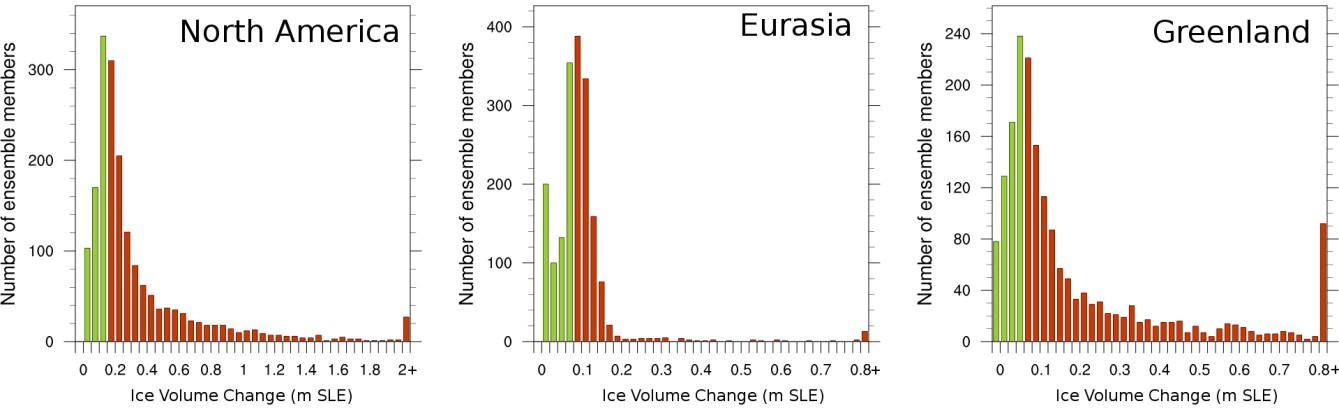

**Figure 10.** The distribution of ice volume change over NA, EA, and Gr, between 1700 and 1980 CE in 2000 ensemble runs. Green bars represent the selected 500 ensemble members, and red bars represent the rest.

5    From here on, we focus on the 500 member sub-ensemble results. Fig. 11 shows the Greenland region ensemble mean thickness and standard deviation at PD. The largest ice thickness changes occur at the southern margins of the Greenland ice sheet. Eastern NA ice expansion is concentrated in the high Arctic (Ellesmere Island and adjacent) where PD ice caps exist.

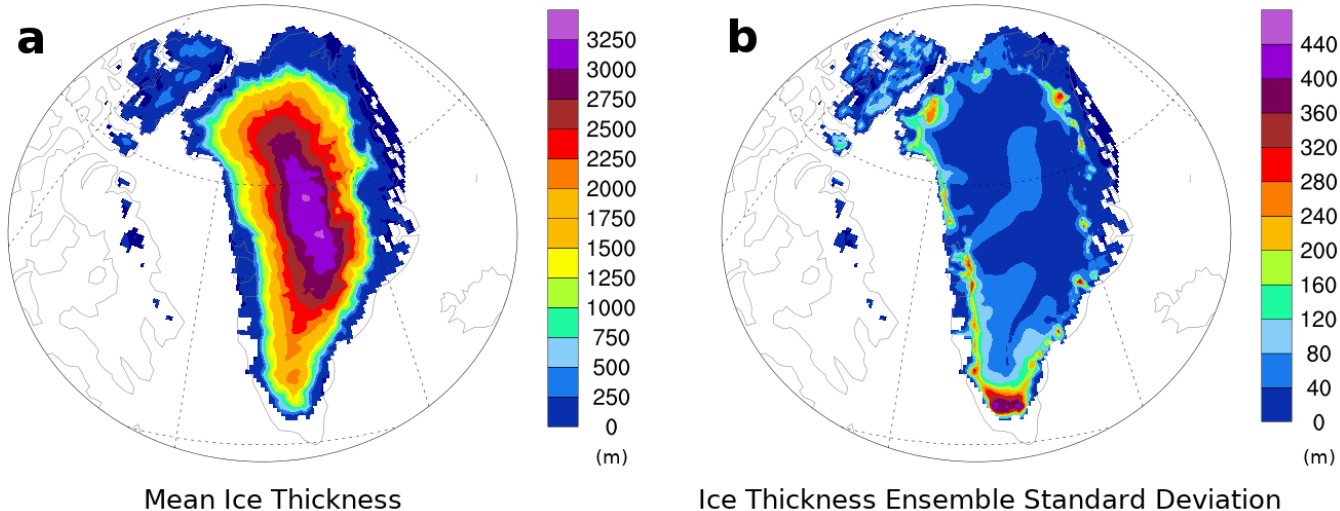

**Figure 11.** Greenland ice thickness ensemble **a**. mean, and **b**. standard deviation at PD.

## 5.1 2 meter temperature and precipitation

The ensemble distribution of the annual mean global T2m anomaly with respect to observations shows that the majority of the ensemble members fall within ± 3°C from observation (grey bars in Fig. 12). However, as ice sheet build up is a function of both temperature and precipitation, most of the warm biased simulations fail to maintain ice-free conditions over NA and EA during PD due to a high winter precipitation bias (Table 3).

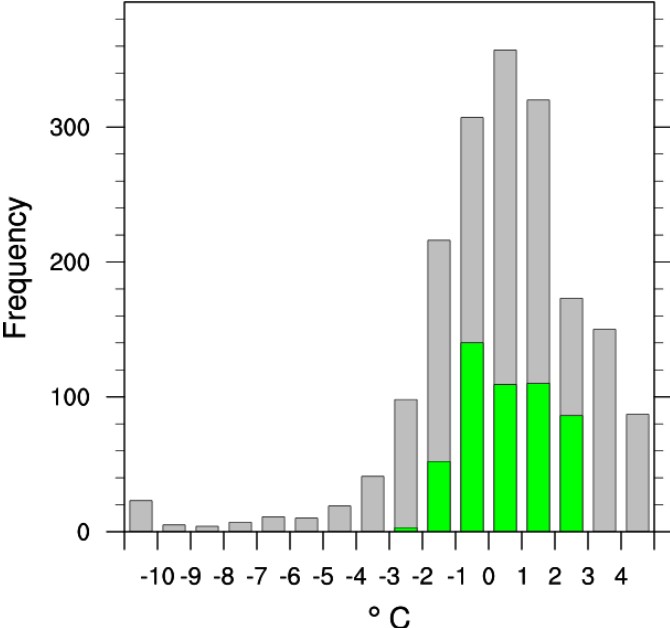

**Figure 12.** The distribution of global annual mean 2 meter temperature difference between ensemble members and observations averaged from 1950 to 1980 CE. Grey and green bars represent the 2000 ensemble members and the top-performing 500 ensemble members respectively.

Our four latitudinal bands defined in Section 4 (Table 3) provide more relevant temperature metrics for NH ice sheet contexts. All four regions have higher ensemble mean seasonal T2m and precipitation compared to observations. However, the observations are covered well within two standard deviations for all regions and temperature is covered within one standard deviation for most regions.

The seasonal cycle provides a partial test of a model's response to orbital forcing on Milankovitch scales. The ensemble mean seasonal cycle (difference between mean summer and mean winter) is within one standard deviation of the reanalysis data for all regions and for both temperature and precipitation (Table 3). Furthermore, aside from NorthEA, the diagnostic regions have a mean difference between summer and winter ensemble temperatures within a degree C of that of the reanalysis climatology.

**Table 3.** The sieved sub-ensemble and observed mean summer and winter 2 meter temperature and precipitation averaged over four latitudinal bands for the 1950-1980 CE interval.

| | | Summer | | Winter | |
|---|---|---|---|---|---|
| | | T2m | Precipitation | T2m | Precipitation |
| Zone | | (°C) | (mm/month) | (°C) | (mm/month) |
| NorthNA | Model Ens. | 6.3±2.3 | 54.3±14.3 | -25.2±3.9 | 20.0±5.4 |
| | Observation | 4.7 | 30.6 | -27.9 | 10.5 |
| SouthNA | Model Ens. | 18.0±2.1 | 85.2±19.7 | -7.7±2.1 | 56.4±5.3 |
| | Observation | 15.0 | 68.5 | -9.7 | 49.4 |
| NorthEA | Model Ens. | 5.6±2.3 | 33.4±5.6 | -8.3±5.5 | 16.7±6.9 |
| | Observation | 3.3 | 27.2 | -14.0 | 8.9 |
| SouthEA | Model Ens. | 14.9±1.8 | 60.7±10.4 | -3.2±2.0 | 55.1±10.4 |
| | Observation | 12.9 | 59.7 | -5.6 | 43.5 |

## 5.2 NH jet-stream

Jet-stream latitude and oscillations have a strong control over storm-tracks and the boundary between polar and subtropical air masses. They are therefore critical factors in controlling where and when an ice sheet margin advances or retreats. Due to the low vertical resolution of ECBilt, we compare the ensemble zonal mean of the 200 hPa zonal wind (as opposed to the more usual 300 hPa diagnostic level) with observations in winter and summer over NA and EA (Fig. 13a). The ensemble shows good agreement in capturing the maximum zonal velocity, but there is a 10 to 15 degree shift northward in the latitude of the jet in both seasons. This is likely due to the reduced temperature gradient between low and high latitude.

We also compare the 30-80°N meridionally averaged meridional wind at 200 hPa of the ensemble mean and the observations to diagnose the Rossby waves. In the summer, the longitudes of the troughs and ridges from the Pacific Ocean to the Atlantic Ocean largely match the reanalysis output within ensemble range (red line in Fig. 13b) with the largest discrepancies over the Eurasian region. During the winter, although the general pattern of the jet-stream oscillations still agrees between the model and the observations (troughs over Eurasia and North America), the mismatch between ensemble members and observations becomes more significant given the higher Rossby wave number of the model ensemble.

## 5.3 Sea ice

High latitude sea ice acts as an insulator for both heat and moisture between the atmosphere and ocean, the two controlling factors for terrestrial ice sheet surface mass-balance. We use the area and minimum latitude extent of the NH sea ice as relevant diagnostics. The general warm bias of the ensemble is reflected in the reduced ice area of the ensemble for both seasons, barely capturing the observed area within the one standard deviation range of the ensemble (Fig. 14). Both March (maximum) and

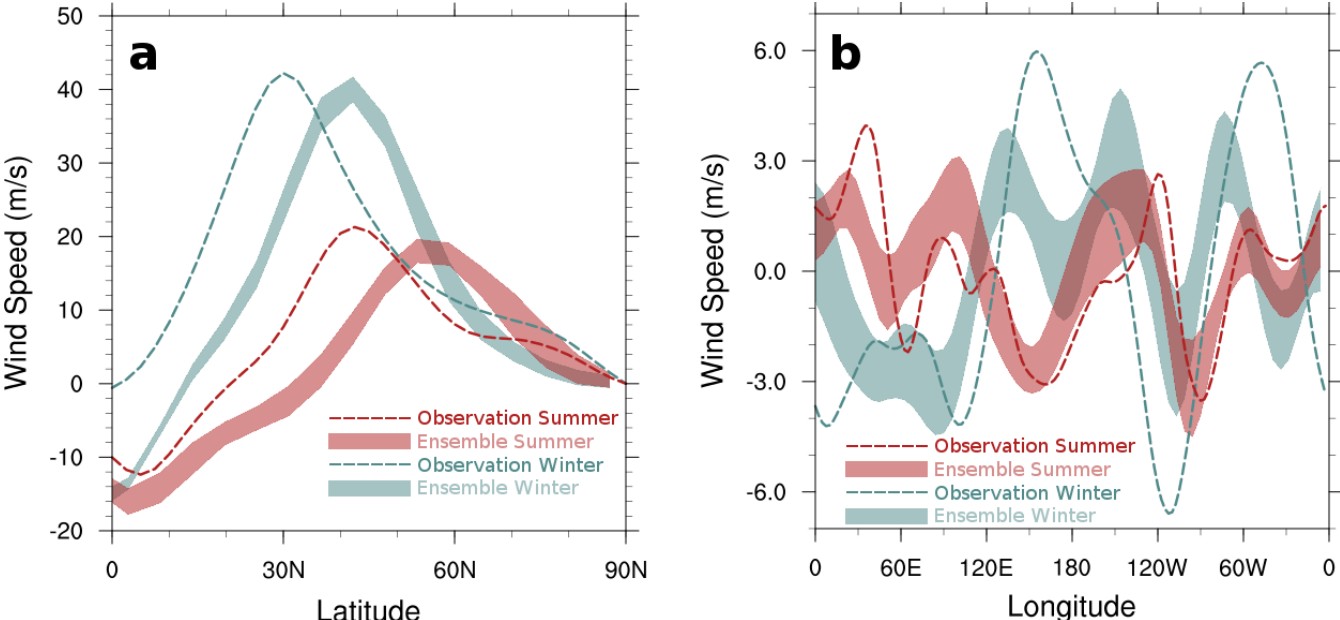

**Figure 13. a**. Zonal average of the zonal component of the 200 hPa wind velocity, and **b**. meridionally average of the meridional component of the 200 hPa wind velocity over the NH. Filled areas show the model ensemble mean and the two standard deviation range. Dashed lines represent observational data. Blue is for winter and red for summer.

September (minimum) NH sea ice areas show gradual decreases in the ensemble mean as the greenhouse gas concentration increases in the model.

10     Our filter condition for our sub-ensemble still permits a wide response of modelled components. For instance, averaging from 1950 to 1980 CE, the Pacific Ocean sea ice shows higher sensitivity to ensemble parameters during its maximum seasonal extent than the Atlantic. The sea ice minimum latitude in the Pacific ranges from 60°N to 45°N (not shown), in comparison to the observed value of 60°N (Walsh et al., 2015).

### 5.4 AMOC

The AMOC transports large amounts of heat and salinity between high and low latitudes. Both paleoclimate proxy records (McManus et al., 2004) and climate model simulations (Liu et al., 2009) show that the AMOC experiences significant changes over a glacial cycle.

    Our "bounding reality" criteria is not met for at least two AMOC features of the sub-ensemble. The PD ensemble mean

5     AMOC strength is weaker than the reanalysis data from European Center of Medium-Range Weather Forecasts (ECMWF ORA-S3, Balmaseda et al., 2008). The temporal mean of the reanalysis data is only captured by the maximum ensemble range (Fig. 15a). The ensemble mean shows a slight increase in AMOC strength from 1965 to 1980 CE, which is not seen in

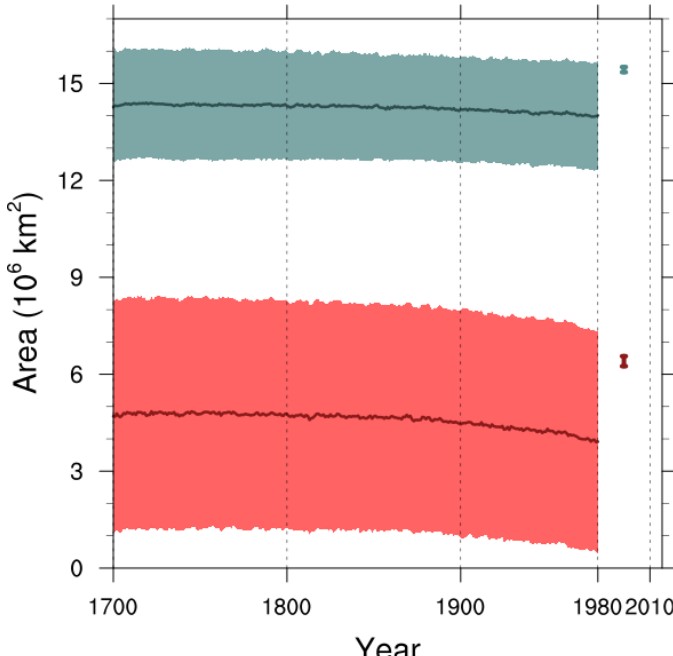

**Figure 14.** Maximum (March: blue) and minimum (September: red) sea ice area ensemble mean ± one standard deviation. The vertical lines represent observational 1981-2010 March and September mean sea ice area within one standard deviation (Walsh et al., 2015).

ORA-S3. As well, the temporal variability of the CLIO AMOC lacks the strong amplitude of the low frequency component of observations (as displayed by the maximum and minimum (time averaged) AMOC strength runs in Fig. 15a).

10      The maximum AMOC stream-function strength is seen around 50°N at 1 km depth. The ensemble variance is also highest in the same region, in addition to 0°latitude at the same depth (Fig. 15b).

## 6   Conclusions

We have coupled an Earth System Model of Intermediate Complexity (LOVECLIM) with a 3D thermomechanical coupled ice sheet systems model (GSM) using LCice 1.0. The coupling efficiently captures most of the relevant feedbacks/interactions

15   between the ice sheet and the atmosphere and ocean models. Our coupled model includes a parameterized sub-shelf melt using upstream ocean vertical temperature profiles, a simple cloud parameterization scheme to improve the radiative forcing representation in the atmosphere model, a dynamical vertical temperature gradient, and a dynamic meltwater runoff routing. We also introduce a new precipitation downscaling scheme that accounts for the change in surface slopes between the coarse resolution climate model grid and the higher resolution ice sheet model grid. Each of the above features has significant impact

on modelled ice thickness (shown directly or via changes in temperature and/or precipitation).

We have presented a set of ensemble parameters to generate an ensemble of runs that "bracket reality" and have shown that each ensemble parameter has a significant impact on modelled PD regional temperatures and/or precipitation. The new

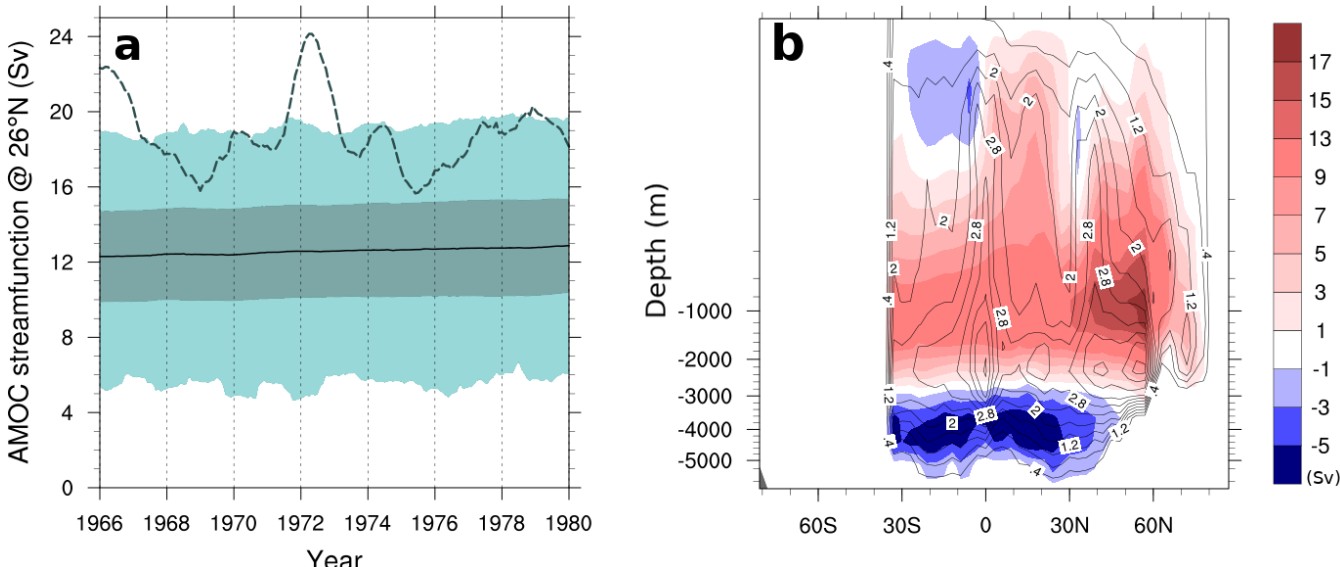

**Figure 15. a**. Maximum AMOC strength at 26°N between 1966 and 1980 CE. Black solid line: ensemble mean; dark blue area: ensemble mean ± one standard deviation; light blue area is bounded by the simulations with the maximum and minimum AMOC strength (time averaged); dashed line: ORA-S3 (Balmaseda et al., 2008). **b**, AMOC stream-function mean (filled colors) and ensemble standard deviation (contour lines).

coupled model was subject to a Latin Hypercube parameter sweep of 2000 ensemble simulations for PD boundary and initial conditions. We extracted a sub-ensemble of 500 model runs according to modelled PD NH ice volume changes. The mean of the sub-ensemble is warm and wet-biased for the NH ice sheet region. However, the model ensemble still brackets reanalysis precipitation and temperature fields within two (ensemble) standard deviations for all regions and within one standard deviation for half of the regions for the case of temperature.

The ensemble's performance at capturing the seasonal cycle is much better. The ensemble mean difference between summer and winter for all four regions is well within one standard deviation of reanalysis values for both temperature and precipitation (and within one degree C for three of the four regions). This provides some confidence that the model responds adequately to orbital forcing (at least for components that operate on sub-annual time-scales).

The "reality bracketing" criterion is not met for certain features of atmospheric circulation (especially winter-time Rossby wave number) and AMOC strength and variance. Another key limitation of LOVECLIM is the inability to change bathymetry and landmask (aside from the parameterized Bering Strait throughflow). The paleoclimate and ice sheet modelling communities would be well served by a modern successor to LOVECLIM for large ensemble glacial cycle time scale contexts that permitted transient changes to bathymetry and landmask.

The coupled model runs at about 1 kyr/day on one core and therefore enables large ensembles of full glacial cycle integrations. As a step towards this, our sub-set of 500 ensemble members is being used for inception and deglaciation ensemble experiments with the coupled model.

*Code and data availability.* LOVECLIM is freely available from http://www.climate.be/modx/index.php?id=81. The GSM will be made publicly available in one to two years (as detailed code documentation and further upgrades progress) as a community model. The LCice 1.0 coupling routines/scripts, modified version of LOVECLIM 1.3, and GSM modules for reading LCice output and computing advective precipitation corrections are freely available at

http://www.physics.mun.ca/~lev/LCice1p0BahadoryTarasov18.tgz.

*Competing interests.* The authors have no competing interests

*Acknowledgements.* The authors thank Heather Andres for editorial help. This paper benefitted from reviews by Dr. Irina Rogozhina and an anonymous reviewer.

This work was supported by a NSERC Discovery Grant (LT), the Canadian Foundation for Innovation (LT), the Atlantic Computational Excellence Network (ACEnet), CREATE, InnovateNL(LT), and the Canada Research Chairs program (LT).

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
