# Peer review of "LCice 1.0: A generalized Ice Sheet Systems Model coupler for LOVECLIM version 1.3: description, sensitivities, and validation with the Glacial Systems Model (GSM version D2017.aug17)"

_Geoscientific Model Development, 2017_

## Short Comment (SC1) · 31 Jan 2018

Dear authors,

in my role as Executive editor of GMD, I would like to bring to your attention our Editorial version 1.1:

http://www.geosci-model-dev.net/8/3487/2015/gmd-8-3487-2015.html

This highlights some requirements of papers published in GMD, which is also available on the GMD website in the 'Manuscript Types' section:

http://www.geoscientific-model-development.net/submission/manuscript_types.html

In particular, please note that for your paper, the following requirement has not been met in the Discussions paper:

- "The main paper must give the model name and version number (or other unique identifier) in the title."

Please add the version numbers for LOVECLIM and GSM to the title upon your revised submission to GMD.

Yours,

Astrid Kerkweg
* * *

---

## Referee Comment (RC1) · Anonymous Referee #1 · 16 Feb 2018

**GENERAL COMMENTS:**

The authors present a new, coupled climate - Northern Hemisphere ice sheet model consisting of LOVECLIM and GSM. The model description focuses on a set of novel, previously often neglected coupling features, and on the effect of these features on the glacial inception (120-106ka).

The conclusion that all of the four new features have significant impacts on the ice sheet evolution - which I assume is the ice sheet evolution during the last glacial inception - is not fully supported by the presented results. While the four new features mentioned in the abstract are 1) dynamic meltwater runoff routing, 2) advective precipitation, 3) a

locally-varying dynamic vertical temperature gradient for lapse-rate corrections, and 4) a sub-shelf melting parameterization, sensitivity tests for the ice sheet evolution during the inception are shown for the inclusion of meltwater runoff (Figure 5), and at least as a snapshot for advective precipitation (Figure 3), but neither for the dynamic lapse-rate correction nor for sub-shelf melting. On the other hand, the sensitivity of the ice sheet evolution to different coupling timesteps and to the inclusion of a cloud parameterization are shown, but not mentioned in the abstract.

While I am a fan of the many sensitivity tests shown, I think that the manuscript would benefit from a more detailed analysis of what exactly causes the differences. For example, why does the changed river runoff lead to enhanced ice buildup during the inception, or how can a larger coupling-timestep cause faster melting?

In the second part of the manuscript, an ensemble of present-day coupled model runs is presented, aimed at "bracketing" the present-day climate (and at reducing the temperature bias of LOVECLIM / page 10 lines 4-5?) by varying ice and snow albedos, initial conditions, and other climate and coupling parameters. A subset of ensemble simulations is then selected based on the simulated 1700-1980 sea-level trend. Although this ensemble of simulations is impressive, it is unclear to me how these large-ensemble sensitivity results presented in Table 2 work together with the first part of the manuscript (sensitivity of simulated glacial inception to new coupling processes). Wouldn't it make more sense to first find a good set of ensemble parameters based on present-day observations, and then try to simulate the last glacial inception?

Since the coupled model is not publicly available, the results are only reproducible for co-operators of the authors. I do think that the description of the four new coupling features (or Section 3 in general) can still be useful for other modelling groups, who are developing or are planning to develop similar model setups (of course, making in particular the coupling-code immediately available would make reusage or development easier - maybe also for the authors, because the description of the coupling in the text might not have to be as detailed and could focus on why a certain method is applied

rather than how exactly that is done). From just the text, it would be challenging if not impossible to reproduce the same coupling schemes (see, e.g., questions below about meltwater runoff).

SPECIFIC COMMENTS:

abstract / page 1, lines 3-4: "... interactions... that are often ignored" Please rephrase, e.g., "neglected" rather than ignored? For example river runoff changes were discussed frequently before, i.e., not ignored, but neglected. Can a "novel downscaling" really be "ignored"?

page 3, lines 5-6: I think it should be pointed out here that, if dust deposition and its surface albedo effect is to be included, then the PDD-approach will have to be replaced or at least modified to include insolation.

page 3, line 6-7: For clarity: The land-sea mask and bathymetry can be changed (see Roche et al., 2007, for LGM bathymetry), but dynamic land-sea mask or bathymetry has not been implemented.

page 3, lines 10-12: I think a list of the other available models, and of the reasons why they could not be used on what architecture would be beneficial (both for the model developers and potential users).

Table 1: Table 1 serves well to illustrate that the four mechanisms discussed here were neglected in many previous continental-scale ice sheet-climate simulations. However, adding "dust" to the table may suggest that this is a complete list of the important coupling mechanisms, which I do not agree with. I would suggest to only include the 4 new features, or to extend the list, including e.g.: stationary wave feedback, vegetation feedbacks, cloud feedbacks, ice sheets or also shelves, grounding line dynamics vs. prescribed ice mask, fixed/variable land-sea mask, and maybe also models used, simulated timeslice or period.

page 4, line 3: Which version of LOVECLIM is used?

page 4, lines 3-4: It should be noted that LOVECL\*IM\* already contains an ice sheet model (AGISM), which, I assume, is not available to the authors?

page 4, line 20: As mentioned above, changing bathymetry seems to be, in principle, possible (Roche et al., 2007), but interactive bathymetry has not been implemented. Or is it clear already that interactive bathymetry will never be possible with CLIO?

page 5, lines 11-12: Since the sub-glacial melting is one of the 4 key features discussed, the parameterization should be described in more detail here.

page 5, lines 12-13: Why is the sub-shelf melting also dependent on proximal subglacial meltwater discharge? I am curious to know what the mechanism behind this assumption is, or if there is observational evidence.

page 5 lines 19-20 / Figure 1: "Figure 1 displays all the fields the coupler transfers..." Figure 1 does not show all the coupling fields and processes. For example, winds, temperature, evaporation, temperature gradient, and ice mask changes are not indicated. Maybe, instead of Figure 1, a box-diagram that really does include all processes would be more accurate, maybe something like Fig. 4 of Roche et al. (2014)? I think such a box-diagram could work very well here, since Section 3 is already organized in "component-coupling-directions", so that arrows in the box-diagram could be named after the subsections in the text.

page 5, line 27: Why does the 100-year-coupling-timestep-run diverge from the others during the melting?

page 6, lines 2-3: What time period is used to compute the fields, only the last year, or for example the last 10 years before the coupling?

page 7, line 5: "The downscaled mean standard deviation..." Mean over several years? Are standard deviations computed from daily means? Or individual timesteps?

page 10, lines 9-11: sub-shelf melting Do I understand correctly that the temperature profiles at the 10 index-sites from Figure 4 are diagnosed at every coupling timestep,

and then prescribed at the proximate shelves? What does "downstream" mean here? Downstream according to the 3D-ocean currents at each coupling timestep? And if so, what happens if the grounding-line / shelf ice moves inland with respect to the CLIO bathymetry? The locations of the index points seem arbitrary, how were they chosen?

page 10, lines 4-5: "We... do not apply a bias correction... and instead examine the extent to which an ensemble parameter sweep can reduce the bias." What is the result of the examination? Can the bias be reduced, and if so, how?

page 11, line 11: "... or preserving the valleys (w2=0), ..." If I understand correctly, the mean topographic height is preserved for w2=0, not the valleys.

page 11, line 17-18: "Any cell in the resulting grid with more than 30% ice coverage is then assumed to be ice covered." I think that this critical ice fraction would also be a good candidate for a sensitivity study. Please briefly discuss possible implications of this choice. I suspect that a critical fraction smaller than (the maybe more natural choice of) 50% facilitates the ice buildup during the inception?

page 11, lines 24-27 (freshwater discharge): Is the runoff flux calculated by the land model affected by the ice cover? What happens to the soil moisture in ice-covered grid cells? What happens in (according to the ECBilt-grid) partially ice-covered grid cells at ice margins? If a grid cell is covered by, say, 40% of ice and is as such defined as ice-covered, is all precipitation onto the land-grid-box blocked, although only 40% of the grid box "sees" precipitation? Are the "LOVECLIM discharge zones" the coloured areas in Figure 4? Shouldn't the coupler compute the freshwater produced in the respective catchment areas that correspond to the discharge zones, and then \*discharge\* that water in the discharge zones? Which mass exactly is conserved? Are the catchment basins and discharge areas adjusted to the changing LOVECLIM topography?

page 11, line 30: "... ice in the simulation including runoff grows faster as it gains more volume." This sounds like a very interesting feedback; why does the runoff enhance ice growth? I suspect a cooling due to a weaker meridional overturning?

page 18, line 10: Which orbital and GHG forcings are used?

Figure 8: The x-Axis label should probably be "ice volume change (m SLE / 280 years)" rather than ice volume.

page 22 line 10: replace "amounts of heat and freshwater" by "amounts of heat and seawater"?

page 22 line 13: "Our bounding reality criteria is not met by at least two AMOC features..." [replace "by" by "for"?]

**TECHNICAL CORRECTIONS:**

p2 l28: "(e.g., Roche et al., 2014)" [brackets, dots]

p2 l31: maybe better: "..., we are also working towards bracketing the effects of these feedbacks." [if not, then "strengths" with s?]

p4 l9: no space after "hours"

p4 l19: describe (no "s")

p7 l6: monthly mean [add -ly]

p11 l6: of all GSM grid cells [no "the"]

p22 l2: no dot after "ensemble."

p22 l4: full-stop missing

Fig 11 caption: Zonal average [no -ly]

Fig 13 caption: remove double brackets around "Balmaseda et al"

p28: please check reference Nikolova et al., 2013 [remove copyright statement]

---

## Referee Comment (RC2) · I. Rogozhina (Referee) · 26 Feb 2018

Review by Irina Rogozhina, MARUM, University of Bremen

The manuscript of Bahadory and Tarasov presents a new coupled climate-ice sheet model, which is capable of time-efficient simulations of entire glacial-interglacial cycles. In order to bridge the low-resolution of the climate model (T21 for the atmosphere) with the higher resolution of the ice sheet model (0.5° x 0.25°) the authors have introduced important tools for the downscaling of climate fields, which are most relevant to the calculation of the ice sheet surface mass balance (i.e., precipitation and near-surface temperature). In addition, the paper presents a self-consistent treatment

of the ocean-driven ice shelf melting, daily temperature standard deviation (DTSD) in the temperature-index method, dynamic routing of supraglacial hydrology and spatially distributed freshwater fluxes and tests the impacts of these model developments in coupled simulations of the glacial inception during Marine Isotope Stage (MIS) 5. Taken together this is a solid contribution to the ongoing work towards the development of comprehensive models for multimillennial climate simulations, and I recommend it for publication after moderate revisions.

I agree with reviewer 1 that the paper will benefit from some restructuring. Indeed, it would make sense to first present the model validation and calibration over the observational period and then move to the evaluation of the impacts of new tools on the modelled climate and ice sheets during MIS5. In addition to the presented analyses it would be interesting to see how the implemented dynamic lapse rate corrections and DTSD improve the model performance compared to the commonly used lapse rates of 6.5 – 7°C/km and uniform DTSD values of 4 - 5°C. In particular, in the light of studies of Erokhina et al. (2017) and Wake and Marshall (2014) it is important to assess, whether these previously inferred dependences of lapse rates and DTSD on background climate are confirmed by long-term transient climate simulations.

I also agree with reviewer 1 that the paper needs a more detailed description of the newly implemented tools. Referring for details to a manuscript, which is not even submitted, is not commonplace. Even though this is a technical paper and thus does not have an appropriate format for extensive discussion and interpretation of the results, some aspects of the development presented in this study require further analysis in order to put these model developments and the resulting model sensitivities into the context of the past climate/ice sheet evolution. For example, the study finds that during MIS5 the Cordilleran ice sheet extended significantly beyond its southernmost marginal positions documented for the later stages of the last glacial cycle (MIS4 and MIS2). If this result is meaningful, glacial imprints of this earlier ice sheet advance should have been preserved until now. Is there any geomorphological/geochronological evidence

confirming such extensive glaciations in North America during MIS5 or it is merely a model artefact? If this is the latter, the impact of dynamic runoff routing seems to amplify an unrealistic ice sheet buildup (Figure 5b) and thus as opposed to one's expectations gives a poor credit to the inclusion of such model development. Could the authors reflect upon this result in the context of the model validation for the glacial inception period?

Minor suggestions:

Page 5, lines 7 – 8: Do the authors mean "ice streams" instead of "ice shelves"?

Page 9, Figure 3: It would be useful to include an absolute ice sheet thickness from the reference experiment in this figure.

Page 13, line 18: distribute → distributed

Page 13, line 26: trigger → are triggered

Page 19, first paragraph: Why are the authors talking about 3 North American ice sheets in the context of the 20th century simulations?

Please, consider including a table with the main model parameters in different sensitivity experiments relative to the reference experiment.

Please, describe in detail how the ice sheet model was initialized.

References

Erokhina, O., Rogozhina, I., Prange, M., Bakker, P., Bernales, J., Paul, A., and Schulz, M. (2017): Dependence of slope lapse rate over the Greenland ice sheet on background climate. Journal of Glaciology, 63, 568-572. Wake L and Marshall S (2015) Assessment of current methods of positive degree-day calculation using in situ observations from glaciated regions. J. Glaciol., 61(226), 329–344 (doi: 10.3189/2015JoG14J116)

---

## Editor Comment (EC1) · D. Goldberg (Editor) · 6 Mar 2018

thanks to the referees for insightful comments – particularly anonymous referee 1 who was very thorough. As editor I am in strong agreement with the comments made by both referees, particularly the disconnect between the aims of the model, and the calibration period.

I would also like to remind the authors that the manuscript as it is does not meet the code availability standards of the journal, as pointed out by ref 1. This is unfortunately a non-starter, as I have confirmed with the senior editors, and the manuscript cannot be accepted for publication until these standards are met.

---

## Author Comment (AC1) · 17 Jul 2018

**LCice 1.0: A generalized Ice Sheet Systems Model coupler for LOVECLIM version 1.3: description, sensitivities, and validation with the Glacial Systems Model (GSM version D2017.aug17)**

Taimaz Bahadory and Lev Tarasov

July 13, 2018

**1** Appreciation**

We thank both reviewers for their thoughtful and detailed reviews.

**2 Response to editor comments**

- **Referee** thanks to the referees for insightful comments particularly anonymous referee 1 who was very thorough. As editor I am in strong agreement with the comments made by both referees, particularly the disconnect between the aims of the model, and the calibration period.
- **Authors** As detailed below in the response to reviewer comments, we have clarified the rational for our choice of present-day for initial ensemble sieving and 110 ka for examples of feedback/interaction impact. Note, we do not describe nor carry out a calibration.
- **Referee** I would also like to remind the authors that the manuscript as it is does not meet the code availability standards of the journal, as pointed out by ref 1. This is unfortunately a non-starter, as I have confirmed with the senior editors, and the manuscript cannot be accepted for publication until these standards are met.
- Authors We have place the coupler archive online as indicated in the revised text with the requisite documentation and example files:

LOVECLIM is freely available from http://www.climate.be/modx/index.php? id=81. The GSM will be made publicly available in one to two years (as detailed code documentation and further upgrades progress) as a community model. The LCice 1.0 coupling routines/scripts, modified version of LOVECLIM 1.3, and GSM modules for reading LCice output and computing advective precipitation corrections are freely available at http://www.physics.mun.ca/~lev/LCice1p0BahadoryTarasov18.tgz. Upon acceptance of the revised paper for GMD (and therefore ensuring all reviewer concerns about code documentation have been addressed), we will place the code archive (or revision thereof) on the zenodo public server and revise the above code availability statement.

The GSM is still in development (especially with respect to internal documentation). Having experienced the frustrations of dealing with poorly documented code, we see no point in providing the GSM code archive. We have changed the title to clarify that this paper is presenting and making available the coupler and revised LOVECLIM code for others to use with their own ice sheet systems models:

LCice 1.0: A generalized Ice Sheet Systems Model coupler for LOVECLIM version 1.3: description, sensitivities, and validation with the Glacial Systems Model (GSM version D2017.aug17)

The above also addresses the editor request to "Please add the version numbers for LOVECLIM and GSM to the title upon your revised submission to GMD".

**3 Discussion with Anonymous Referee 1**

- Referee The authors present a new, coupled climate Northern Hemisphere ice sheet model consisting of LOVECLIM and GSM. The model description focuses on a set of novel, previously often neglected coupling features, and on the effect of these features on the glacial inception (120-106ka). The conclusion that all of the four new features have significant impacts on the ice sheet evolution - which I assume is the ice sheet evolution during the last glacial inception - is not fully supported by the presented results.
- **Authors** We added the cloud parameterization into the list, and now show the impact of each on either ice thickness at 110 ka or the temporal evolution of ice volume during the inception. The plots show that all features have significant impacts. The five features included in the coupling, and the sections and figures discussing the impact of each are as follows:
  - Vertical temperature gradient: section 3.1.1, Fig. 3.c showing the ice thickness difference between the dynamic and the fixed at 6.5 K/km.
  - Advective precipitation downscaling: section 3.1.2, Fig. 4 showing North America (NA) ice thickness difference at 110 ka between simulations with and without advective precipitation method.
  - Sub-shelf melt: section 3.2, Fig. 5 showing ice thickness difference between PD ocean temperature run, fixed ocean temperature at -2 °C, and ocean layers temperature averaged run, and the Control run (dynamic ocean temperature).
  - Freshwater discharge: section 3.4.1, Fig. 7 showing total ice volume evolution at glacial inception with and without dynamic meltwater routing, and NA ice thickness difference at 110 ka with and without runoff routing.
  - Cloud radiation parameterization: section 4.5, Fig. 9 showing total ice volume in SLE at the last glacial inception with the cloud parameterization and with the PD cloud cover forcing.

- Referee While the four new features mentioned in the abstract are 1) dynamic meltwater runoff routing, 2) advective precipitation, 3) a varying dynamic vertical temperature gradient for lapse-rate corrections, and 4) a sub-shelf melting parameterization, sensitivity tests for the ice sheet evolution during the inception are shown for the inclusion of meltwater runoff (Figure 5), and at least as a snapshot for advective precipitation (Figure 3), but neither for the dynamic lapse-rate correction nor for sub-shelf melting.
- Authors For the "vertical temperature lapse-rate" part, we added a new figure (Fig. 3) and the following paragraph at the end of section "3.1.1 Vertical temperature gradient":

Fig. 3.a and 3.b show the present-day vertical T2m lapse-rate calculated by the coupler for summer and winter. The derived lapse-rate has strong spatial and temporal variation over NA and Greenland. The impact of this variation is shown in Fig. 3.c. Starting from the same 110 ka configuration, the difference in ice thickness after 2 kyr between a dynamic temperature lapse-rate run and a control run (default LOVECLIM parameters) with 6.5 K/km lapse rate can reach over 1 km.

Evaluating the appropriateness of our vertical temperature downscaling approach is difficult, especially when considering glacial/interglacial changes. Using a global climate model (CCSM3), Erokhina et al. (2017) found significantly larger surface slope lapse-rate values over the Greenland ice sheet during LGM compared to preindustrial values (February mean increase of about 3.7 K/km and about 0.9 K/km for July). In contrast, our T2m mean LGM lapse-rate over Greenland is 0.8K/kmstronger for February and 0.2K/km weaker for July compared to that of PD. However, neither lapse rate is *apriori* an accurate choice for vertical downscaling. A need remains for a multi-resolution modelling study to compare a "true" downscaling vertical temperature gradient with the various possible lapse-rates that can be derived from a single resolution model.

We also added figure 5 and the following paragraph into the sub-shelf melt section (3.2 Ocean to ice: sub-shelf melt):

To test the impact of this regional disaggregation of ocean temperatures, we generated three test cases: ocean temperature forcing set to PD value, to -2 °C, and set to the contemporaneous average across the above index sites. Starting from a 110 ka restart, all three options have local ice thickness differences greater than 1 km after 2 kyr compared to that with the standard coupling.

**Referee** On the other hand, the sensitivity of the ice sheet evolution to different coupling timesteps and to the inclusion of a cloud parameterization are shown, but not mentioned in the abstract.**

Authors Both are technical issues that we therefore judge do not warrant inclusion in the abstract. We have added the "Dynamic parameterized radiative cloud effect as a function of humidity" as the fifth feature into the introduction. We also have added the following to the introduction concerning coupling time-steps:

We also examined sensitivity to coupling time-step by setting up three similar simulations with different coupling time-steps (100, 20, and 10 years).

- **Referee** While I am a fan of the many sensitivity tests shown, I think that the manuscript would benefit from a more detailed analysis of what exactly causes the differences. For example, why does the changed river runoff lead to enhanced ice buildup during the inception, or how can a larger coupling-timestep cause faster melting?
- **Authors** We disagree. For most cases, the answers are already self evident (eg large differences in lapse rates causing large temperature differences over regions with high elevation gradients) or else too complex to decipher for this paper. The point of the sensitivity analysis is to show that each of these coupling components have significant impacts on model results.

The one exception is the case of river runoff, for which we think the ice growth is mostly due to weakening of AMOC, and the associated drop in the summer temperature (figure not shown). We have added a line to mention AMOC weakening at the end of section "3.4.1 Freshwater Discharge":

Compared to the control run (no dynamic drainage routing), the AMOC strength drops by 15%, and the sea ice extent shows an increase of 5% in winter and  $\sim 15\%$  in summer (not shown). The combination of these two yields a cooler summer in the run with dynamic drainage routing, hence less ice sheet melt.

- Referee In the second part of the manuscript, an ensemble of present-day coupled model runs is presented, aimed at "bracketing" the present-day climate (and at reducing the temperature bias of LOVECLIM / page 10 lines 4-5?) by varying ice and snow albedos, initial conditions, and other climate and coupling parameters. A subset of ensemble simulations is then selected based on the simulated 1700-1980 sea-level trend. Although this ensemble of simulations is impressive, it is unclear to me how these large ensemble sensitivity results presented in Table 2 work together with the first part of the manuscript (sensitivity of simulated glacial inception to new coupling processes). Wouldn't it make more sense to first find a good set of ensemble parameters based on present-day observations, and then try to simulate the last glacial inception?
- Authors We are actually doing what the reviewer suggests. We carry out a 2000 member grand ensemble for present-day and use that to select 500 parameter vectors for the glacial inception ensemble which will be the focus of an upcoming submission to Climates of the Past. The confusion appears to be mis-interpretation of the role of the 110 ka example impacts. Their purpose is not meant as an attempt "to try to simulate the last glacial inception". Instead, they are meant to show that each of our featured coupling components has a potentially significant impact for glacial contexts. We believe it's clearer for the reader to have this presented together with the description of the process/coupling component. In the next section, Table 2 then shows significant sensitivity of model runs over the historical interval to each of our selected ensemble parameters. Having thus justified our choice of parameters, we then cary out the initial grand ensemble.

The following text has been added to the introduction to better clarify our logic:

Next, we describe the coupling schemes between the ice sheet model and the atmosphere and the ocean models in section 3. In this section, we use the last glacial inception timeframe (120 - 110 ka) to show that inclusion of each process coupling scheme can have significant impact on the evolution of major NH ice sheets. In section 4, we introduce our chosen set of ensemble parameters for the coupled model.

In order to justify this choice of ensemble parameters, we examine the sensitivity of the coupled model to changes in each parameter for PD climate. Then we sieve the ensemble parameter set using our coupled model with historical/PD initial and boundary conditions via a comparison against observational/reanalysis data.

Also the caption of Table 2 is updated as follow:

Ensemble parameters that are varied in the historical transient simulation ensemble. Column 2: the distribution of parameter values versus their range for each parameter. Column 3: change in 1950-1980 mean summer 2 meter temperature and winter precipitation over four selected regions when each parameter is varied independently from its minimum to its maximum value. In each sensitivity run, all the other parameters are fixed to LOVECLIM default values, with spin-up length: 4000 years, simple upscaling method, default LOVECLIM cloud radiative forcing, start-year: 1500 AD, and dynamic vertical temperature lapse-rate.

- **Referee** Since the coupled model is not publicly available, the results are only reproducible for cooperators of the authors. I do think that the description of the four new coupling features (or Section 3 in general) can still be useful for other modelling groups, who are developing or are planning to develop similar model setups (of course, making in particular the coupling-code immediately available would make reusage or development easier - maybe also for the authors, because the description of the coupling in the text might not have to be as detailed and could focus on why a certain method is applied rather than how exactly that is done). From just the text, it would be challenging if not impossible to reproduce the same coupling schemes (see, e.g., questions below about meltwater runoff).
- Authors We already stated that the coupling code is freely available upon request. However, given the above comments, the modified LOVECLIM, coupling code, and scripts have been placed online as a tarball as detailed above in response to the editor comments.

SPECIFIC COMMENTS:

- Referee abstract / page 1, lines 3-4: "... interactions... that are often ignored" Please rephrase, e.g., "neglected" rather than ignored? For example river runoff changes were discussed frequently before, i.e., not ignored, but neglected. Can a "novel downscaling" really be "ignored"?
- Authors Fixed.
- **Referee** page 3, lines 5-6: I think it should be pointed out here that, if dust deposition and its surface albedo effect is to be included, then the PDD-approach will have to be replaced or at least modified to include insolation.
- Authors We already state "First, the dust cycle and its impact on atmospheric radiative balance and ice surface albedo (and therefore surface mass balance) awaits future work".
- **Referee** page 3, line 6-7: For clarity: The land-sea mask and bathymetry can be changed (see Roche et al., 2007, for LGM bathymetry), but dynamic land-sea mask or bathymetry has not been implemented.

- Authors We changed the text to: "Second, the LOVECLIM ocean component does not handle changing bathymetry and landmask over a transient run."
- **Referee** page 3, lines 10-12: I think a list of the other available models, and of the reasons why they could not be used on what architecture would be beneficial (both for the model developers and potential users).
- Authors We added the following list to include the models:
  - SPEEDO: compilation error using PGI and Intel compilers.
  - FOAM (v. 1.5): no dynamic sea ice model; compilation error using PGI and Intel compilers.
  - OSUVic (v. 2.8): compilation error.
  - SIRO-Mk3L (v. 1.2): compilation error using PGI, Intel, and GCC compilers; problem accessing fftw library.
- Referee Table 1: Table 1 serves well to illustrate that the four mechanisms discussed here were neglected in many previous continental-scale ice sheet-climate simulations. However, adding "dust" to the table may suggest that this is a complete list of the important coupling mechanisms, which I do not agree with. I would suggest to only include the 4 new features, or to extend the list, including e.g.: stationary wave feedback, vegetation feedbacks, cloud feedbacks, ice sheets or also shelves, grounding line dynamics vs. prescribed ice mask, fixed/variable land-sea mask, and maybe also models used, simulated timeslice or period.
- Authors Yup, this is an incomplete list and a complete list would likely lead to arguments over semantics.. We've now changed the caption to address this and better convey what we had in mind:

Feedbacks/interactions sporadically included in previous studies between the ice sheet model and the rest of the climate system, compared to the current study. None include changes to land mask and bathymetry except for parameterized Bering Strait throughflow.

- **Referee** page 4, line 3: Which version of LOVECLIM is used?
- Authors Version is added: 1.3
- **Referee** page 4, lines 3-4: It should be noted that LOVECLIM already contains an ice sheet model (AGISM), which, I assume, is not available to the authors?
- Authors Yes, it is not freely available. Furthermore the GSM includes a number of processes not in the AGISM (visco-elastic earth rheology for GIA, surface drainage solver, subgrid ice mass-balance/flow, permafrost,...)
- **Referee** page 4, line 20: As mentioned above, changing bathymetry seems to be, in principle, possible (Roche et al., 2007), but interactive bathymetry has not been implemented. Or is it clear already that interactive bathymetry will never be possible with CLIO?
- Authors We only know that LOVECLIM currently doesn't handle interactive bathymetry and landmask. We have therefore changed the indicated line to:

A major limitation of this model (and challenge for many GCMs) for paleoclimate studies is that the bathymetry and land mask can't be changed during a transient run (specifically, there is no available nor described implementation that can do so).

- **Referee** page 5, lines 11-12: Since the sub-glacial melting is one of the 4 key features discussed, the parameterization should be described in more detail here.
- Authors We modified the text as follow:

The melt is proportional to the water temperature to the power 1.6 and to proximal sub-glacial meltwater discharge following the Greenland fjord modelling results of Xu et al. (2013). We also impose a quadratic dependence on ice thickness to concentrate sub-shelf melt near deep grounding lines in accord with the results of process modelling (e.g., Jacobs et al., 1992).

- **Referee** page 5, lines 12-13: Why is the sub-shelf melting also dependent on proximal subglacial meltwater discharge? I am curious to know what the mechanism behind this assumption is, or if there is observational evidence.
- **Authors** This is a basic feature of sub-shelf melt physics (eg cf the citation in the above description of the new sub-shelf melt parameterization). Sub-shelf melt is highly enhanced by the development of a convective cell to continually entrain new water to the ice basal surface and the convection rate is dependent on basal freshwater discharge at the grounding line.
- Referee page 5 lines 19-20 / Figure 1: "Figure 1 displays all the fields the coupler transfers..." Figure 1 does not show all the coupling fields and processes. For example, winds, temperature, evaporation, temperature gradient, and ice mask changes are not indicated. Maybe, instead of Figure 1, a box-diagram that really does include all processes would be more accurate, maybe something like Fig. 4 of Roche et al. (2014)? I think such a box-diagram could work very well here, since Section 3 is already organized in "component-coupling-directions", so that arrows in the box-diagram could be named after the subsections in the text.
- Authors Replaced with a new diagram showing the coupling directions.
- **Referee** page 5, line 27: Why does the 100-year-coupling-timestep-run diverge from the others during the melting?
- Authors We added the following text to section 3:

With identical boundary and initial conditions for all three simulations, runs with 10 and 20 year coupling steps have less than a maximum of 3% difference in ice volume (Fig. 2). The 100-year coupling-step run (red line in Fig. 2), however, strongly diverges from the other two during the retreat phase. This ice volume divergence is mostly due to a thinner ice in North America (NA) and Eurasia (EA), and a less southern extent of the NA ice sheet. A weaker response with longer coupling time-steps is expected given the delay in updating climate and ice boundary conditions for the GSM and LOVECLIM respective.

**Referee** page 6, lines 2-3: What time period is used to compute the fields, only the last year, or for example the last 10 years before the coupling?

Authors We fixed it by adding "averaged over the last 10 years".

**Referee** page 7, line 5: "The downscaled mean standard deviation..." Mean over several years? Are standard deviations computed from daily means? Or individual timesteps?

Authors We now state in the text:

The downscaled standard deviation of temperature (using 4-hourly ECBilt data for each month averaged over the last 10 years of each coupling time-step) is used to compute monthly Positive Degree Days, with the usual assumption of a Gaussian distribution around the monthly mean. This is opposed to the traditional practice of assuming a constant value, usually between 5°C and 7°C.

Referee page 10, lines 9-11: sub-shelf melting Do I understand correctly that the temperature profiles at the 10 index-sites from Figure 4 are diagnosed at every coupling timestep, and then prescribed at the proximate shelves? What does "downstream" mean here? Downstream according to the 3D-ocean currents at each coupling timestep? And if so, what happens if the grounding-line / shelf ice moves inland with respect to the CLIO bathymetry? The locations of the index points seem arbitrary, how were they chosen?

Authors We have modified the text as follow:

Sub ice shelf melt is a challenge for paleo coupled ice sheet climate modelling given the dependence on unresolved basin-scale circulation. As a first order approximation, we assume that upstream ocean temperature at the same depth corresponds to the local sub-shelf temperature. To facilitate fast and simplified coupling, given the complexity of ocean grids in most ocean general circulation models, we only extract upstream ocean temperature vertical profiles from LOVECLIM at the end of each coupling time-step for a number of chosen index sites as indicated in Fig. 5 and use these for downstream marine sectors. We selected these sites (7 over NA+Greenland and 4 over EA) by examining PD ocean temperature climatologies from CLIO (at various depths) while taking into account ocean currents. Current site selection was predicated on the constant bathymetry and land mask of CLIO and would need updating for a model with dynamic landmask/bathymetry. The downstream masks for the profile sites extend onto land where applicable when grounding line retreat beyond the fixed ocean mask of CLIO (i.e. onto the land mask) is possible.

To test the impact of this regional disaggregation of ocean temperatures, we generated three test cases: ocean temperature forcing set to PD value, to -2°C, and set to the contemporaneous average across the above index sites. Starting from a 110 ka restart, all three options have local ice thickness differences greater than 1 km after 2 kyr compared to that with the standard coupling.

- **Referee** page 10, lines 4-5: "We... do not apply a bias correction... and instead examine the extent to which an ensemble parameter sweep can reduce the bias." What is the result of the examination? Can the bias be reduced, and if so, how?
- Authors We examined individual runs biases and compared them with the control run bias (with parameters stated in the Table 2's caption). The results are summarized and included in the paper as follow:

The control run (with all LOVECLIM parameters set to their default values, and other coupling parameters as stated in the caption of Table 2) shows the highest temperature bias in the "Southern NA" region ( $\sim 5^{\circ}$ C), with slightly colder temperatures in the "North NA" ( $\sim 1^{\circ}$ C). The temperature bias over EA is less significant, and is also less latitude dependent (both "Northern EA" and "Southern EA" biased by  $< 2^{\circ}$ C). A reduction in regional temperature and precipitation bias is observed in various members of our later introduced ensemble of simulations for PD. The regional temperature and precipitation bias relative to observed (Table 2) over NA and EA can reach zero for some ensemble members for both summer and winter. Although there is no individual run with zero bias in all the regions, a number of selected runs show reduced temperature biases (between -1°C and 1°C) in all the four regions compared to that of the control run.

**Referee** page 11, line 11: "... or preserving the valleys (w2=0), ..." If I understand correctly, the mean topographic height is preserved for w2=0, not the valleys.

**Authors fixed**

- Referee page 11, line 17-18: "Any cell in the resulting grid with more than 30% ice coverage is then assumed to be ice covered." I ' think that this critical ice fraction would also be a good candidate for a sensitivity study. Please briefly discuss possible implications of this choice. I suspect that a critical fraction smaller than (the maybe more natural choice of) 50% facilitates the ice buildup during the inception?
- Authors We agree the threshold could be a candidate for sensitivity analysis and have added the following to the end of "Ice-mask" section:

Our choice of a 30% threshold (as opposed to say 50%) was motivated by the following logic. For any atmospheric grid cell covering an ice margin segment, the temperature passed to the GSM should most importantly reflect ice covered boundary conditions local to the ablations zone of the ice sheet. Allowance for subgrid advection of warmer air masses from adjacent ice-free land somewhat tempers this logic. Given potentially significant impacts on critical ablation temperatures and therefore ice-sheet mass-balance, this ice-fraction threshold deserves a sensitivity analysis (in future work).

- Referee page 11, lines 24-27 (freshwater discharge): Is the runoff flux calculated by the land model affected by the ice cover? What happens to the soil moisture in ice-covered grid cells? What happens in (according to the ECBilt-grid) partially ice-covered grid cells at ice margins? If a grid cell is covered by, say, 40% of ice and is as such defined as ice-covered, is all precipitation onto the land-grid-box blocked, although only 40% of the grid box "sees" precipitation? Are the "LOVECLIM discharge zones" the coloured areas in Figure 4? Shouldn't the coupler compute the freshwater produced in the respective catchment areas that correspond to the discharge zones, and then \*discharge\* that water in the discharge zones? Which mass exactly is conserved? Are the catchment basins and discharge areas adjusted to the changing LOVECLIM topography?
- Authors We have changed the title of the relevant subsection to partly answer the reviewer: "Topographicallyself-consistent and mass conserving freshwater discharge". We have added the following to the

"Freshwater discharge" section to explain in more detail how both the LOVECLIM and the GSM drainage systems are connected through the coupler.

As the GSM self-consistently computes surface drainage (while conserving mass) for the evolving topography while LOVECLIM surface drainage is hard-coded for PD topography, precipitation within LOVECLIM is masked out where covered by the GSM grid. The coupler then distributes GSM freshwater ocean discharge to corresponding LOVECLIM ocean discharge grid cells. Where the GSM grid edge is terrestrial, the GSM discharge is added to the corresponding LOVECLIM grid cell for internal runoff routing. As topographic gradient changes from Glacial Isostatic Adjustment (GIA) are small outside of the GSM grid, this scheme should give results close to what would be achieved with a drainage solver using a topography globally subject to GIA.

We also added a new plot to a supplement to clarify the connection between GSM and LOVE-CLIM discharge sites, and how on-land routing is processed in the coupler. Here are the associated paragraphs:

LOVECLIM has 26 predefined water discharge zones globally (colored cells in Fig. 1 (only the NH)) abutting continental margins within which the runoff flux calculated by the land model is uniformly distributed. The coupler then receives the GSM discharge at either continental margins or the terrestrial GSM grid boundaries (black lines in Fig. 1).

In the case of the continental margin, all the GSM drainage in regions bounded by same-color brackets are directed into LOVECLIM drainage cells with the similar color as the brackets. For instance, the GSM drainage south of Alaska between the two purple brackets is dumped into the four LOVECLIM purple cells in the same region. The LOVECLIM drainage module uniformly redistributes this discharge across the region.

Over the terrestrial GSM grid boundaries, the GSM runoff between same-color brackets are redirected to LOVECLIM drainage cells based on PD drainage maps. The GSM drainage in these regions is added to the runoff calculated by LOVECLIM from the non-GSM covered regions. For instance, southern Europe runoff between the olive-green brackets are redirected into the Mediterranean, in addition to the runoff calculated by LOVECLIM from southern Europe and northern Africa.

For the south-eastern Eurasian margin of the GSM grid, the GSM runoff is directed to the Pacific ocean for the following reason. Most of the drainage in this largely dry region is northward except for that of the Caspian Sea watershed. As this sea is absent in LOVECLIM and all its mass-loss is evaporative, prevailing westerly winds would dictate predominantly eastward transport of moisture.

- Referee page 11, line 30: "... ice in the simulation including runoff grows faster as it gains more volume." This sounds like a very interesting feedback; why does the runoff enhance ice growth? I suspect a cooling due to a weaker meridional overturning?
- Authors We looked at the climatic fields in two runs (with and without dynamic drainage routing) to investigate the main differences in the climatic fields. Our findings are summarized as follow:

Compared to the control run (no dynamic drainage routing), the AMOC strength drops by 15%, and the sea ice extent shows an increase of 5% in winter and  $\sim 15\%$  in summer (not shown). The combination of these two yields a cooler summer in the run with dynamic drainage routing, hence less ice sheet melt.

**Referee** page 18, line 10: Which orbital and GHG forcings are used?

Authors We modified the text as follow:

All simulations are spun up using transient forcings (orbital (Berger 1978) and GHG Law Dome (for recent CO2 data) and Dome C (Etheridge et al., 1998; Monnin et al., 2001) (for pre-industrial to 5 ka)) for 3000 to 5000 years without the GSM coupled, followed by a transient coupled run ending at year 1980.

- **Referee** Figure 8: The x-Axis label should probably be "ice volume change (m SLE / 280 years)" rather than ice volume.
- Authors Fixed
- Referee page 22 line 10: replace "amounts of heat and freshwater" by "amounts of heat and seawater"?

Authors Fixed

**Referee** page 22 line 13: "Our bounding reality criteria is not met by at least two AMOC features..." [replace "by" by "for"?]

**Authors Fixed**

TECHNICAL CORRECTIONS:

Referee p2 l28: "(e.g., Roche et al., 2014)" [brackets, dots]

p2 l31: maybe better: "..., we are also working towards bracketing the effects of these feedbacks." [if not, then "strengths" with s?]

- p4 l9: no space after "hours"
- p4 l19: describe (no "s")
- p7 l6: monthly mean [add -ly]

p11 l6: of all GSM grid cells [no "the"]

p22 l2: no dot after "ensemble."

p22 l4: full-stop missing Fig 11 caption: Zonal average [no -ly]

Fig 13 caption: remove double brackets around "Balmaseda et al"

p28: please check reference Nikolova et al., 2013 [remove copyright statement]

Authors All fixed in the text.

**4 Discussion with Dr. Irina Rogozhina**

**Referee** The manuscript of Bahadory and Tarasov presents a new coupled climate-ice sheet model, which is capable of time-efficient simulations of entire glacial-interglacial cycles. In order to bridge the low-resolution of the climate model (T21 for the atmosphere) with the higher resolution of the ice sheet model  $(0.5 \circ x 0.25 \circ)$  the authors have introduced important tools for the downscaling of climate fields, which are most relevant to the calculation of the ice sheet surface mass balance (i.e., precipitation and near-surface temperature). In addition, the paper presents a self-consistent treatment of the ocean-driven ice shelf melting, daily temperature standard deviation (DTSD) in the temperature-index method, dynamic routing of supraglacial hydrology and spatially distributed freshwater fluxes and tests the impacts of these model developments in coupled simulations of the glacial inception during Marine Isotope Stage (MIS) 5. Taken together this is a solid contribution to the ongoing work towards the development of comprehensive models for multimillennial climate simulations, and I recommend it for publication after moderate revisions.

I agree with reviewer 1 that the paper will benefit from some restructuring. Indeed, it would make sense to first present the model validation and calibration over the observational period and then move to the evaluation of the impacts of new tools on the modelled climate and ice sheets during MIS5.

Authors cf response to similar reviewer #1 comment and the cited revised text (repeated below) that clarifies our logic for the current structure. We choose to follow the logic (and our own work-flow) of how we and we suspect some other modellers add processes. We are not going to waste computational resources in doing a grand ensemble with a new set of parameters/processes until we've tested whether each parameter/process has a potentially significant impact for the context of interest. We believe it's best to have the logic of the paper reflect the logic of the model development.

Next, we describe the coupling schemes between the ice sheet model and the atmosphere and the ocean models in section 3. In this section, we use the last glacial inception timeframe (120 - 110 ka) to show that inclusion of each process coupling scheme can have significant impact on the evolution of major NH ice sheets. In section 4, we introduce our chosen set of ensemble parameters for the coupled model. In order to justify this choice of ensemble parameters, we examine the sensitivity of the coupled model to changes in each parameter for PD climate. Then we sieve the ensemble parameter set using our coupled model with historical/PD initial and boundary conditions via a comparison against observational/reanalysis data.

- Referee In addition to the presented analyses it would be interesting to see how the implemented dynamic lapse rate corrections and DTSD improve the model performance compared to the commonly used lapse rates of 6.5 to 7 K/km and uniform DTSD values of 4 - 5 °C. In particular, in the light of studies of Erokhina et al. (2017) and Wake and Marshall (2014) it is important to assess, whether these previously inferred dependences of lapse rates and DTSD on background climate are confirmed by long-term transient climate simulations.
- Authors We added a new figure to the paper (Fig. 3) with the following caption to show the present-day seasonal vertical temperature lapse-rate spatial distribution over NA, and also the ice-thickness

difference between a dynamic temperature lapse-rate run and the control run (fixed lapse-rate at 6.5 K/km):

Figure 3 caption:

Vertical temperature lapse-rate calculated by the coupler at PD over NA in **a**. February, and **b**. July. **c**. shows the ice thickness difference between dynamic and constant 6.5 K/km lapse rate (control) runs after running for 2 kyr, starting from the same 110 ka configuration. Black contours show the ice thickness in the control run. Thick black and green contours show the ice margin in the control and dynamic lapse-rate run, respectively.

The differences are discussed in more detail in the last paragraph of section 3.1.1 as given in the response to reviewer 1 above.

As for documenting the sensitivity to chosen DTSD, we respectfully disagree. Sensitivity to the scalar values of DTSD has previously been documented and the cited Wake and Marshall paper have made clear the lack of observational support for a constant DTSD. Extracting DTSD from an EMIC is easy, so this is an easy upgrade to a model and should therefore become standard practice for those coupled models using PDD variant schemes. The paper focus is feedbacks/interactions between ice and climate often ignored, not the impact of every feature of the coupled model. The paper is more than big enough as is and if we were to document the sensitivity to every feature of the coupled GSM/EMIC not present in most other models the paper would be twice as long and therefore too long for anyone to want to read.

- **Referee** I also agree with reviewer 1 that the paper needs a more detailed description of the newly implemented tools.
- Authors To calculate the "near-surface vertical temperature lapse-rate", we used a method similar to that of Roche et al. (2014), described as follows:

In each LOVECLIM grid cell, the coupler first determines the highest and lowest elevations from the GSM topography constrained by the cell's boundary. Next, the T2m for these two elevations is calculated using the inherited scheme from the LOVECLIM atmospheric model (Roche et al., 2014). The resulting temperatures and elevation difference between the two points is then used to calculate the temperature lapse-rate in that LOVECLIM grid cell.

For the freshwater discharge, we added a new paragraph to the "Freshwater discharge" section to explain in more detail how both the LOVECLIM and the GSM drainage systems are connected through the coupler. We also added a new plot and associated text to the supplement to clarify the connection between GSM and LOVECLIM discharge sites, and how on-land routing is processed in the coupler. All these changes are detailed in the response to reviewer 1 above.

**Referee** Referring for details to a manuscript, which is not even submitted, is not commonplace. Even though this is a technical paper and thus does not have an appropriate format for extensive discussion and interpretation of the results, some aspects of the development presented in this

study require further analysis in order to put these model developments and the resulting model sensitivities into the context of the past climate/ice sheet evolution. For example, the study finds that during MIS5 the Cordilleran ice sheet extended significantly beyond its southernmost marginal positions documented for the later stages of the last glacial cycle (MIS4 and MIS2). If this result is meaningful, glacial imprints of this earlier ice sheet advance should have been preserved until now. Is there any geomorphological/geochronological evidence confirming such extensive glaciations in North America during MIS5 or it is merely a model artefact? If this is the latter, the impact of dynamic runoff routing seems to amplify an unrealistic ice sheet buildup (Figure 5b) and thus as opposed to one's expectations gives a poor credit to the inclusion of such model development. Could the authors reflect upon this result in the context of the model validation for the glacial inception period?

Authors We respectfully disagree. This paper is not about analysing glacial inception simulations (which would be inappropriate for GMD), the glacial inception period is used to show the impact of implementing new processes in the coupled model. The sensitivity examples have not been chosen for best fit to observations (the inception ensemble was far from complete when this manuscript was submitted). To repeat our logic, the additions all represent self-evident improved physical representation and/or self-consistency. The tests are to show these all have significant impact.

The Cordilleran ice sheet extension issue and other geological evidence matches/mismatches will be discussed in a separate paper specifically focused on the glacial inception period.

Minor suggestions:

- Referee Page 5, lines 7 8: Do the authors mean "ice streams" instead of "ice shelves"
- Authors No, correct as stated (and thus why the word "crude" is used).
- **Referee** Page 9, Figure 3: It would be useful to include an absolute ice sheet thickness from the reference experiment in this figure.
- Authors Ice thickness contours added.
- **Referee** Page 13, line 18: distribute > distributed
- Authors Disagree
- **Referee** Page 13, line 26: trigger -> are triggered
- Authors fixed
- **Referee** Page 19, first paragraph: Why are the authors talking about 3 North American ice sheets in the context of the 20th century simulations?
- **Authors** We used the present-day simulated Northern Hemisphere ice sheet growth to sieve out parameter vectors with major surface mass-balance biases. To make this clearer in the text we've inserted:

Our focus on coupled ice and climate and our choice to avoid bias corrections led to a trial criteria based on ice volume changes (between 1700 and 1980 CE). Therefore we used the PD simulated NH ice sheet growth to sieve out parameter vectors with major surface mass-balance biases.

- **Referee** Please, consider including a table with the main model parameters in different sensitivity experiments relative to the reference experiment.
- Authors Table 2 already provides this.
- Referee Please, describe in detail how the ice sheet model was initialized.

[revised manuscript text omitted]
 down-scaling downscaling scheme which accounts for wind direction-velocity and topographic slopes.
- 3. Dynamic meltwater routing.
- 4. Parameterized sub-shelf melt using An efficient scheme to extract approximate lat/long gridded ocean temperature fields from LOVECLIM ocean temperature profiles for sub ice shelf melt computation

Table 1 compares the feedbacks interactions between ice sheets and climate models only infrequently included in previous coupled modelling studies to this one. There are two main feedbacks interactions yet to be implemented. First, the dust cycle and its impact on atmospheric radiative balance and ice surface albedo (and therefore surface mass balance) awaits future work. Second, the LOVECLIM ocean component ean does not handle changing bathymetry and landmask over a transient run. It does

20 have a parameterized Bering Strait throughflow which permits shutdown of throughflow when local water depth approaches zero.

The climate model used in the coupled model needs Climate models used for glacial cycle contexts need to be fast enough to simulate tens of thousands of years in a reasonable time interval, while sufficiently complex to include all important climate dynamics. We tested every freely available fast model that included ocean, atmosphere and dynamical sea ice components, and

25 found a number of published models to be numerically unstable or otherwise unable to run or port. The only stable model with all these components was LOVECLIM. The other models tested and associated porting failures are:

**SPEEDO** : compilation error using PGI and Intel compilers.

FOAM (v. 1.5) : no dynamic sea ice model; compilation error using PGI and Intel compilers.

OSUVic (v. 2.8) : compilation error.

30 CSIRO-Mk3L (v. 1.2) : compilation error using PGI, Intel, and GCC compilers; problem accessing fftw library.

**Table 1.** Feedbacks/interactions sporadically included in previous studies between the ice sheet model and the rest of the climate system, compared to the current study. None include changes to land mask and bathymetry except for parameterized Bering Strait throughflow.

| Source       | Advective     | Dynamic vertical     | Dynamic meltwater | Sub-shelf    | Dust         |
|--------------|---------------|----------------------|-------------------|--------------|--------------|
|              | precipitation | temperature gradient | runoff routing    | melt         | deposition   |
| ?            | $\bigotimes$  | $\bigotimes$         | $\bigotimes$      | $\otimes$    | $\otimes$    |
| ?            | $\bigotimes$  | $\bigotimes$         | $\bigotimes$      | $\bigotimes$ | $\bigotimes$ |
| ?            | $\otimes$     | $\bigotimes$         | $\odot$           | $\otimes$    | $\otimes$    |
| ?            | $\bigotimes$  | $\bigotimes$         | $\bigotimes$      | $\bigotimes$ | $\bigotimes$ |
| ?            | $\otimes$     | $\bigotimes$         | $\otimes$         | $\otimes$    | $\odot$      |
| ?            | $\otimes$     | $\odot$              | $\otimes$         | $\odot$      | $\otimes$    |
| ?            | $\bigotimes$  | $\bigotimes$         | $\bigotimes$      | $\otimes$    | $\otimes$    |
| Current work | $\odot$       | $\odot$              | $\odot$           | $\odot$      | $\bigotimes$ |

The paper is structured as follows. We first introduce the models in section 2. Next, we describe the coupling schemes between the ice sheet model and the atmosphere and the ocean models in section 3and their impact on ice sheet evolution during. In this section, we use the last glacial inception timeframe (120 - 110 ka) to show that inclusion of each process coupling scheme can have significant impact on the evolution of major NH ice sheets. In section 4, we introduce the our

5 chosen set of ensemble parameters for the coupled modeland the ranges used for each. We. In order to justify this choice of ensemble parameters, we examine the sensitivity of the coupled model to changes in each parameter for Present-Day (PD) PD climate. Then we test sieve the ensemble parameter set using our coupled model with historical/PD initial and boundary conditions and compare the results with via a comparison against observational/reanalyzed reanalysis data.

**2 Models**

15

**10 2.1 LOVECLIM**

[revised manuscript text omitted]

response with longer coupling time-steps is expected given the delay in updating climate and ice boundary conditions for the GSM and LOVECLIM respectively. Given these results, we choose <del>20-years as the optimum 20</del> years as the coupling step for all of our ensemble simulations (in part given the not insignificant overhead with the coupler as currently coded/scripted).